# On the Limitations of Temperature Scaling for Distributions with Overlaps

**Muthu Chidambaram**
Department of Computer Science
Duke University
muthu@cs.duke.edu

**Rong Ge**
Department of Computer Science
Duke University
rongge@cs.duke.edu

## Abstract

Despite the impressive generalization capabilities of deep neural networks, they have been repeatedly shown to be overconfident when they are wrong. Fixing this issue is known as model calibration, and has consequently received much attention in the form of modified training schemes and post-training calibration procedures such as temperature scaling. While temperature scaling is frequently used because of its simplicity, it is often outperformed by modified training schemes. In this work, we identify a specific bottleneck for the performance of temperature scaling. We show that for empirical risk minimizers for a general set of distributions in which the supports of classes have overlaps, the performance of temperature scaling degrades with the amount of overlap between classes, and asymptotically becomes no better than random when there are a large number of classes. On the other hand, we prove that optimizing a modified form of the empirical risk induced by the Mixup data augmentation technique can in fact lead to reasonably good calibration performance, showing that training-time calibration may be necessary in some situations. We also verify that our theoretical results reflect practice by showing that Mixup significantly outperforms empirical risk minimization (with respect to multiple calibration metrics) on image classification benchmarks with class overlaps introduced in the form of label noise.

## 1 Introduction

The past decade has seen a rapid increase in the prevalence of deep learning models across a variety of applications, in large part due to their impressive predictive accuracy on unseen test data. However, as these models begin to be applied to critical applications such as predicting credit risk (Clements et al., 2020), diagnosing medical conditions (Esteva et al., 2017; 2021; Elmarakeby et al., 2021), and autonomous driving (Bojarski et al., 2016; Grigorescu et al., 2020), it is crucial that the models are not only accurate but also predict with appropriate levels of uncertainty.

In the context of classification, a model with appropriate uncertainty would be correct with a probability that is similar to its predicted confidence – for example, among samples on which the model predicts a class with 90% confidence, around 90% of them should indeed be the predicted class (a more formal definition is provided in Equation (2.1)).

As a concrete (but highly simplified) example, consider the case of applying a deep learning model for predicting whether a patient has a life-threatening illness (Jiang et al., 2012). In this situation, suppose our model classifies the patient as not having the illness but does so with high confidence. A physician using this model for their assessments may then incorrectly diagnose the patient (with potentially grave consequences). On the other hand, if the model had lower confidence in this incorrect prediction, a physician may be more likely to do further assessments.

Obtaining such models with good predictive uncertainty is the problem of *model calibration*, and has seen a flurry of recent work in the context of training deep learning models (Guo et al., 2017; Thulasidasan et al., 2019; Ovadia et al., 2019; Wen et al., 2020; Minderer et al., 2021). In particular, Guo et al. (2017) showed that the simple technique of *temperature scaling* – which introduces only a single parameter to "dampen" the logits of a trained model (defined formally in Section 2) – is a powerful procedure for calibrating deep learning models.

Temperature scaling falls under the class of post-training calibration techniques, which are attractive due to the fact that they can be applied to a black box model without requiring any kind of retraining. However, several empirical works have shown that temperature scaling alone can be outperformed by training-time modifications such as data augmentation (Thulasidasan et al., 2019; Müller et al., 2020) and regularized loss functions (Kumar et al., 2018; Mukhoti et al., 2020).

In this work, we try to understand these empirical observations theoretically by addressing the following question:

> *Can we identify reasonable conditions on the data distribution for which temperature scaling provably fails to achieve good calibration, but training-time modifications still succeed?*

## 1.1 Main Contributions and Outline

We answer this question in the affirmative, showing that temperature scaling cannot handle data distributions with certain class overlap properties, while training-time modifications can. We first define the notions of calibration and temperature scaling relevant to our work in Section 2, and then motivate conditions on the models we consider in our theory in Section 3.1. Namely, we focus on models that interpolate the training data (i.e. achieve zero training error) and satisfy a local Lipschitz-like condition. We also introduce the Mixup (Zhang et al., 2017) data augmentation, as well as a modification of it necessary for our theoretical results in Section 3.2.

After establishing the necessary background, we show in our main results of Section 4 that for classification tasks in which the supports of different classes have overlaps, the performance of temperature scaling degrades with the amount of overlap between classes, while the performance of our modified Mixup procedure remains robust to such overlaps. The key idea behind our theory is that *training-time calibration techniques can significantly constrain model behavior in regions away from the training data points*, and this notion can be extended beyond our Mixup analysis to other augmentation techniques.

Lastly, in Section 5 we show that our theoretical results accurately reflect practice by considering both synthetic data and image classification benchmarks. In Section 5.1 we show that the performance of empirical risk minimization (ERM) combined with temperature scaling quickly degrades on a simple 2-class high-dimensional Gaussian dataset as we increase the overlap between the two Gaussians. Similarly, in Section 5.2 we show that the same phenomenon occurs on standard datasets when we increase the amount of label noise in the data (thereby increasing class overlaps).

## 1.2 Related Work

**Calibration in deep learning.** The calibration of deep learning models has received significant attention in recent years, largely stemming from the work of Guo et al. (2017) which empirically showed that modern, overparameterized models can have poor predictive uncertainty. Follow-up works (Thulasidasan et al., 2019; Rahaman & Thiery, 2021; Wen et al., 2021) have supported these findings, although the recent work of Minderer et al. (2021) showed that current state-of-the-art architectures can be better calibrated than the previous generation of models.

**Methods for improving calibration.** Many different methods have been proposed for improving calibration, including: logit rescaling (Guo et al., 2017), data augmentation (Thulasidasan et al., 2019; Müller et al., 2020), ensembling (Lakshminarayanan et al., 2017; Wen et al., 2020), and modified loss functions (Kumar et al., 2018; Wang et al., 2021). The logit rescaling methods, namely temperature scaling and its variants (Kull et al., 2019; Ding et al., 2020), constitute perhaps the most applied calibration techniques, since they can be used on any trained model with the introduction of only a few extra parameters (see Section 2). However, we show in this work that this kind of post-training calibration can be insufficient for some data distributions, which can in fact require data augmentation/modified loss functions to achieve good calibration. We focus particularly on Mixup (Zhang et al., 2017) data augmentation, whose theoretical benefits for calibration were recently studied by Zhang et al. (2021) in the context of linear models and Gaussian data. Our results provide a complementary perspective to this prior work, as we address a much broader class of models and a different class of data distributions.

## 2 THEORETICAL PRELIMINARIES

**Notation.** We use $[k]$ to denote $\{1, 2, ..., k\}$ for a positive integer $k$. We consider $k$-class classification and use $\mathcal{X}$ to denote a dataset of $N$ points $(x_i, y_i)$ sampled from a distribution $\pi(X, Y)$ whose support $\text{supp}(\pi)$ is contained in $\mathbb{R}^d \times [k]$. We use $\pi_X$ and $\pi_Y$ to denote the respective marginal distributions of $\pi$, and use $\pi_y$ to denote the conditional distribution $\pi(X \mid Y = y)$. We use $d(A, B)$ to denote the Euclidean distance between two sets $A, B \subset \mathbb{R}^d$, $d_{\text{KL}}(\pi_1, \pi_2)$ to denote the KL divergence between two distributions $\pi_1$ and $\pi_2$, and $\mu_d$ for the Lebesgue measure on $\mathbb{R}^d$. For a function $g : \mathbb{R}^d \to \mathbb{R}^k$, we use $g^i$ to denote the $i^{\text{th}}$ coordinate function of $g$. Lastly, we use $\phi(\cdot)$ to denote the softmax function, i.e. $\phi^i(g(x)) = \exp(g^i(x)) / \sum_{j \in [k]} \exp(g^j(x))$. In everything that follows, we assume both $N$ and $k$ are sufficiently large, and that $N = \Omega(\text{poly}(k))$ for some large degree polynomial $\text{poly}(k)$.

**Calibration.** In a classification setting, we say that a model $g$ is *calibrated* with respect to the ground-truth probability distribution $\pi$ if the following conditional probability condition holds:

$$\mathbb{P}(Y \mid \phi(g(X))) = \phi(g(X)) \tag{2.1}$$

Equation (2.1) captures the earlier mentioned intuition that, when our model predicts the probability distribution $\phi(g(X))$ over the classes $[k]$, the true probability distribution for the classes is also $\phi(g(X))$. It is straightforward to translate Equation (2.1) into a notion of miscalibration by considering the expectation of the norm of the difference between the left and right-hand sides. In practice, however, it is common in multi-class classification to focus only on calibration with respect to the top predicted class $\text{argmax}_y \phi^y(g(X))$. This leads to the top-class expected calibration error (ECE), which we define simply as ECE below:

$$\text{ECE}(g) = \mathbb{E}_{(X,Y) \sim \pi} \left[ \left\| \mathbb{E}[Y \in \underset{y \in [k]}{\text{argmax}} \, \phi^y(g(X)) \mid \underset{y \in [k]}{\max} \, \phi^y(g(X)) = p^*] - p^* \right\| \right] \tag{2.2}$$

Top-class ECE is only one of several common notions used to measure calibration, and is known to have several theoretical and empirical drawbacks (Błasiok et al., 2023). In our theoretical work, we opt to instead work with the expected KL divergence $\mathbb{E}_{X \sim \pi_X}[d_{\text{KL}}(\pi(Y \mid X), \cdot)]$ as our notion of calibration error. Note that such a notion is difficult to estimate in reality as we do not know $\pi(Y \mid X)$. However, we will consider cases where the post-training calibration procedure has access to $\pi(Y \mid X)$. In such cases expected KL divergence is a more accurate characterization as minimizing it implies the full calibration condition of Equation (2.1), while minimizing Equation (2.2) does not (we will only be calibrated with respect to the top class).

With a notion of miscalibration in hand, we consider methods to improve the calibration of a trained model $g$. One of the most popular (and simplest) approaches is *temperature scaling* (Guo et al., 2017), which consists of introducing a single parameter $T$ that is used to scale the outputs of $g$. The value of $T$ is obtained by optimizing the negative log-likelihood on a calibration dataset $\mathcal{X}_{\text{cal}}$:

$$T = \underset{\hat{T} \in (0, \infty)}{\text{argmin}} -\frac{1}{|\mathcal{X}_{\text{cal}}|} \sum_{(x_i, y_i) \in \mathcal{X}_{\text{cal}}} \log \phi^{y_i}(g(x_i)/\hat{T}) \tag{2.3}$$

For our results, we will in fact consider an even more powerful (and impractical) form of temperature scaling in which we allow access to the ground-truth distribution $\pi$:

$$T = \underset{\hat{T} \in (0, \infty)}{\text{argmin}} \, \mathbb{E}_{X \sim \pi_X} \left[ d_{\text{KL}}(\pi(Y \mid X), \phi^Y(g(X)/\hat{T})) \right] \tag{2.4}$$

We will henceforth refer to an optimally temperature-scaled model with respect to Equation (2.4) as $g_T$. We will show in Section 4 that even when we allow this "oracle" temperature scaling, we cannot hope to calibrate models $g$ that satisfy some empirically-observed regularity properties.

## 3 TRAINING APPROACHES

### 3.1 EMPIRICAL RISK MINIMIZATION

In practice, models are often trained via empirical risk minimization (ERM). Due to the large number of parameters in modern models, training often leads to an *interpolator* that is very confident on every training data point $(x_i, y_i) \in \mathcal{X}$, as we formalize below:

**Definition 3.1.** [ERM Interpolator] For a dataset $\mathcal{X}$, we say that a model $g$ is an ERM interpolator if for every $(x_i, y_i) \in \mathcal{X}$ there exists a universal constant $C_i$ such that:

$$\min_{s \neq y_i} g^{y_i}(x_i) - g^s(x_i) > \log k \quad \text{and} \quad \max_{r,s \neq y_i} g^s(x_i) - g^r(x_i) < C_i \tag{3.1}$$

Equation (3.1) is slightly stronger than directly assuming $\phi^{y_i}(g(x_i)) \approx 1$; however, this type of significant logit separation is commonly observed in practice. Indeed, to further justify Equation (3.1) we train ResNeXt-50 (Xie et al., 2016) models on CIFAR-10, CIFAR-100, and SVHN and examine the means of the max and second max logit over the training data; results are shown in Table 1.

| Logit | CIFAR-10 | CIFAR-100 | SVHN |
|--------|----------|-----------|---------|
| Max | 14.5755 | 18.6198 | 11.5285 |
| Second Max | 0.3650 | 5.4575 | -4.1904 |

Table 1: Comparison of means of max and second max logit over different datasets, with ResNeXt-50 models trained as described in Section 5.

In addition to the interpolation condition of Definition 3.1, we also constrain our attention to ERM models $g$ that satisfy a mild local-Lipschitz-like condition.

**Definition 3.2.** [$\gamma$-Regular] For a point $(x_i, y_i) \in \mathcal{X}$, letting $L$ be a universal constant, we define:

$$\mathcal{B}_\gamma(x_i) = \{x \in \mathbb{R}^d : \|x_i - x\| \leq \gamma\} \tag{3.2}$$

$$\mathcal{G}_\gamma(x_i) = \{x \in \mathbb{R}^d : |g^{y_i}(x_i) - g^{y_i}(x)| \leq L\gamma\} \tag{3.3}$$

We say that a model $g$ is $\gamma$-regular over a set $U$ if there exists a class $y \in [k]$ and $\Theta(\pi_X(U)N)$ points $(x_i, y) \in \mathcal{X}$ with $x_i \in U$ such that $\pi_y(X \in \mathcal{G}_\gamma(x_i) \mid X \in \mathcal{B}_\gamma(x_i)) \geq 1 - O(1/k)$.

Basically, Definition 3.2 codifies the idea that the model logit $g^{y_i}$ does not change much in a small enough neighborhood of each $x_i$ over a given set $U$, with high probability. Experiments providing empirical justification for this assumption are provided in Appendix B.1. We will show in Theorem 4.5 that satisfying Definitions 3.1 and 3.2 is sufficient for poor calibration for a wide class of data distributions, even when using temperature scaling with access to the ground truth distribution oracle.

### 3.2 MIXUP

In contrast, we can show that if we consider models minimizing a Mixup-like training objective instead of the usual negative log-likelihood in Equation (2.3), we can prevent major calibration issues.

Let $\mathcal{D}_\lambda$ denote a continuous distribution supported on $[0, 1]$ and let $z_{i,j}(\lambda) = \lambda x_i + (1 - \lambda)x_j$ (using $z_{i,j}$ when $\lambda$ is clear from context) where $(x_i, y_i), (x_j, y_j) \in \mathcal{X}$. Then we may define the empirical Mixup cross-entropy $J_{\text{mix}}(g, \mathcal{X}, \mathcal{D}_\lambda)$ as:

$$J_{\text{mix}}(g, \mathcal{X}, \mathcal{D}_\lambda) = -\frac{1}{N^2} \sum_{i \in [N]} \sum_{j \in [N]} \mathbb{E}_{\lambda \sim \mathcal{D}_\lambda} \left[\lambda \log \phi^{y_i}(g(z_{i,j})) + (1 - \lambda) \log \phi^{y_j}(g(z_{i,j}))\right] \tag{3.4}$$

Essentially, minimizing Equation (3.4) forces a model to linearly interpolate between its predictions $\phi^{y_i}(g(x_i))$ and $\phi^{y_j}(g(x_j))$ over the line segment connecting the points $x_i$ and $x_j$. This already provides some intuition for why Mixup-optimal models can be better calibrated: their predictions can change quickly as one moves away from the training data, avoiding issues that stem from $\gamma$-regularity of the logits combined with interpolation.

However, the line segment constraints of $J_{mix}(g, \mathcal{X}, \mathcal{D}_\lambda)$ will not be enough to make this intuition rigorous when the data is in $\mathbb{R}^d$ with $d > 1$, since in this case line segments are measure zero sets with respect to $\mu_d$. We will thus augment Mixup to work with convex combinations of $d + 1$ points as opposed to two, and refer to this new objective as $d$-**Mixup**.

In generalizing from Mixup to $d$-Mixup, it is helpful from a theoretical standpoint to constrain the set of allowed mixings $\mathcal{M}_d(\mathcal{X}) \subset [N]^{d+1}$. We will consider only mixing points at most some constant distance away from one another, and we will also preclude mixing points that are too highly

correlated.[1] The precise definition of $\mathcal{M}_d(\mathcal{X})$ can be found in Definition A.1 of Appendix A.2; we omit it here due to its technical nature.

Now let $\mathcal{D}_{\lambda,d}$ denote a continuous distribution supported on the $d$-dimensional probability simplex $\Delta^d \subset \mathbb{R}^{d+1}$. Defining $z_\sigma(\lambda) = \sum_{j \in [d+1]} \lambda_j x_{\sigma_j}$ for $\lambda \in \operatorname{supp}(\mathcal{D}_{\lambda,d})$ and $\sigma \in \mathcal{M}_d(\mathcal{X})$, we can define the empirical $d$-Mixup cross-entropy $J_{\mathrm{mix},d}(g, \mathcal{X}, \mathcal{D}_{\lambda,d})$:

$$J_{\mathrm{mix},d}(g, \mathcal{X}, \mathcal{D}_{\lambda,d}) = -\frac{1}{|\mathcal{M}_d(\mathcal{X})|} \sum_{\sigma \in \mathcal{M}_d(\mathcal{X})} \mathbb{E}_{\lambda \sim \mathcal{D}_{\lambda,d}} \left[ \sum_{j \in [d+1]} \lambda_j \log \phi^{y_{\sigma_j}}(g(z_\sigma(\lambda))) \right] \quad (3.5)$$

We will henceforth use $\mathcal{X}_{\mathrm{mix},d}$ to denote the set of all $z_\sigma$. The main benefit of introducing the set $\mathcal{M}_d(\mathcal{X})$ instead of just generalizing Equation (3.4) to mixing over $[N]^{d+1}$ is that it allows us to use a reparameterization trick with which we can characterize the $d$-Mixup optimal prediction at every mixed point $z_\sigma$. We state only an informal version of this result below and defer a formal statement and proof to Appendix A.2.

**Lemma 3.3.** [Informal Optimality Lemma] Every $g^* \in \operatorname{arginf}_g J_{\mathrm{mix},d}(g, \mathcal{X}, \mathcal{D}_{\lambda,d})$ (where the arginf is over all extended $\mathbb{R}^d$-valued functions) satisfies $\phi^y(g^*(z)) = \xi_y(z)/\sum_{s \in [k]} \xi_s(z)$ for almost every $z_\sigma \in \mathcal{X}_{\mathrm{mix},d}$, where $\xi_y(z)$ corresponds to the expected weight of class $y$ points over all mixing sets $\sigma \in \mathcal{M}_d(\mathcal{X})$ from which we can obtain $z$.

We note that this lemma is analogous to Lemma 2.3 in the work of Chidambaram et al. (2021), but avoids restrictions on the function class being considered and is non-asymptotic. Since we can characterize optimal predictions over $\mathcal{X}_{\mathrm{mix},d}$, we can define $d$-Mixup interpolators as follows.

**Definition 3.4.** [$d$-Mixup Interpolator] For a dataset $\mathcal{X}$, we say that $g$ is a $d$-Mixup interpolator if $\phi^y(g(z)) = \phi^y(g^*(z)) \pm O(1/k)$ for almost every $z \in \mathcal{X}_{\mathrm{mix},d}$ and $y \in [k]$, with $g^* \in \operatorname{arginf}_g J_{\mathrm{mix},d}(g, \mathcal{X}, \mathcal{D}_{\lambda,d})$.

In Theorem 4.6, we will show that $d$-Mixup interpolators can achieve good calibration on a subclass of distributions for which ERM interpolators perform poorly.

**Remark 3.5.** In practice it is unreasonable to mix $d + 1$ points when $d$ is large. However, we conjecture that due to the structure of practical models (i.e. neural networks), even mixing two points as in traditional Mixup is sufficient for achieving neighborhood constraints like those induced by $d$-Mixup. We introduce $d$-Mixup because we make no such structural assumptions in our theory.

## 4 MAIN THEORETICAL RESULTS

In this section, we show that even for simple data distributions, ERM interpolators can prevent temperature scaling from producing well-calibrated models, while modifications in the training process (Mixup) can potentially address this issue. Prior to proving our main results, we begin first with a 1-dimensional example that contains the key ideas of our analysis. The full proofs of all results in this section can be found in Appendix A.

### 4.1 WARM-UP: A SIMPLE 1-D EXAMPLE

**Definition 4.1.** [$\alpha$-Overlapping Intervals] Let $\tau(y)$ denote the parity of a nonnegative integer $y$ and let $\beta_y = \lfloor (y-1)/2 \rfloor k + \alpha \tau(y-1)$ for $y \in [k]$, where $\alpha \in [0,1]$ is a parameter of the distribution. Then we define $\pi(X, Y)$ to be the distribution on $\mathbb{R} \times [k]$ such that $\pi_Y$ is uniform over $[k]$ and $\pi(X \mid Y = y)$ is uniform over $[\beta_y, \beta_y + 1]$.

Definition 4.1 corresponds to a distribution in which consecutive class-conditional densities are supported on overlapping intervals (whose overlap is determined by the parameter $\alpha$) with a spacing of $k$ between each pair of classes (see Figure 1). The spacing of $k$ between pairs of classes is introduced only to simplify the $d$-Mixup analysis; it is not necessary for proving the negative results regarding ERM interpolators, and will not feature when we generalize to Definition 4.4.

---

[1]This means we do not mix points with themselves in $d$-Mixup; however, when $\pi_X$ has a density, this makes little difference since we can mix in a neighborhood of any point.

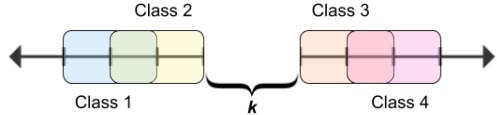

Figure 1: Visualization of Definition 4.1 for the case $k = 4$.

Our first result shows that when considering ERM interpolators for distributions of the type in Definition 4.1, so long as the interpolators satisfy $\gamma$-regularity for sufficiently large $\gamma$, they will be poorly calibrated in the overlapping regions of the data distribution.

**Proposition 4.2.** Let $\mathcal{X}$ consist of $N$ i.i.d. draws from the distribution $\pi$ specified in Definition 4.1, with a parameter $\alpha$. Then with probability at least $1 - k \exp(-\Omega(N/k))$ over the randomness of $\mathcal{X}$, the set $\mathcal{S}$ of all models $g$ that are ERM interpolators for $\mathcal{X}$ and $k/(4N)$-regular over each overlapping region in $\text{supp}(\pi_X)$ is non-empty (in fact, uncountable). Furthermore, the predictive distribution $\hat{\pi}_T(Y \mid X) = \phi^Y(g_T(X))$ of the optimally temperature-scaled model $g_T$ for any $g \in \mathcal{S}$ satisfies:

$$\mathbb{E}_{X \sim \pi_X} \left[ d_{\text{KL}}(\pi(Y \mid X), \hat{\pi}_T(Y \mid X)) \right] \geq \Theta((1 - \alpha - 1/k) \log k) \tag{4.1}$$

Thus, for $\alpha = O(1)$, even with oracle temperature scaling every $g \in \mathcal{S}$ is asymptotically no better than random. In contrast, as the separation $\alpha \to 1$, the bound in Equation (4.1) becomes vacuous.

**Proof Sketch.** We can show that $\mathcal{S}$ is non-trivial using Chernoff bound arguments, and then use $k/(4N)$-regularity to show that there is a significant fraction of $\text{supp}(\pi_X)$ on which every $g \in \mathcal{S}$ predicts incorrect probabilities. The key idea is then that temperature scaling will only improve incorrect predictions for ERM interpolators to uniformly random (i.e. $1/k$), whereas the correct prediction in an overlapping region is $1/2$ for each of the overlapping classes.

On the other hand, for $d$-Mixup, each point in the overlapping regions can be obtained as a mixture of points from the overlapping classes, so we will have non-trivial probabilities for both classes.

**Proposition 4.3.** Let $\mathcal{X}$ be as in Proposition 4.2 and $p(k)$ denote a polynomial in $k$ of degree at least one. Then taking $\mathcal{D}_{\lambda,1}$ to be uniform, *every* 1-Mixup interpolator $g$ for $\mathcal{X}$ with the property that $\phi^y(g(x)) \leq 1 - \Omega(1/p(k))$ for every $x \in \text{supp}(\pi_X) \setminus \mathcal{X}_{\text{mix},1}$ and $y \in [k]$ satisfies with probability at least $1 - k^3 \exp(-\Omega(N/k^3))$:

$$\mathbb{E}_{X \sim \pi_X} \left[ d_{\text{KL}}(\pi(Y \mid X), \hat{\pi}(Y \mid X)) \right] \leq \Theta(1) \tag{4.2}$$

Note that this result is independent of the separation parameter $\alpha$.

**Proof Sketch.** We can show with high probability that $\mathcal{X}_{\text{mix},1}$ covers most of $\text{supp}(\pi_X)$ uniformly, and then we can use Lemma 3.3 to precisely characterize the 1-Mixup predictions over $\mathcal{X}_{\text{mix},1}$.

The added stipulation that $\phi^y(g(x)) \leq 1 - \Omega(1/\text{poly}(k))$ is necessary, since we cannot hope to prove an upper bound if $g$ is allowed to behave arbitrarily on $\text{supp}(\pi_X) \setminus \mathcal{X}_{\text{mix},1}$, and we also expect this in practice due to the regularity of models considered. A key takeaway from Proposition 4.3 is that the upper bound on the Mixup error we obtain is independent of the parameter $\alpha$ of our distribution; we will see that this is also the case in practice in Section 5.

## 4.2 GENERALIZING TO HIGHER DIMENSIONS

By extending the idea of overlapping regions in $\text{supp}(\pi_X)$ from our 1-D example, we can generalize the failure of $\gamma$-regular ERM interpolators to higher-dimensional distributions.

**Definition 4.4.** [General Data Distribution] Given a parameter $\alpha \in [0, 1]$, we define $\pi$ to be any distribution whose support is contained in $\mathbb{R}^d \times [k]$ satisfying the following constraints:

1. (Classes are roughly balanced) $\pi_Y(Y = y) = \Theta(1/k)$.

2. (Constant class overlaps) Letting $M$ denote a nonnegative integer constant, there exist $\Theta(k)$ classes $y$ for which there are classes $s_1(y), s_2(y), ..., s_m(y)$ for some $1 \leq m \leq M$ with $\pi_y(\text{supp}(\pi_y) \cap \text{supp}(\pi_{s_i(y)})) \geq 1 - \alpha$, and all other $s' \in [k]$ satisfy $\pi_X(\text{supp}(\pi_y) \cap \text{supp}(\pi_{s'})) = 0$.

3. (Overlap density is proportional to measure) $\pi_y(X \in A) = \Theta(\mu_d(A))$ and $\pi_{s_i(y)}(X \in A) = \Theta(\mu_d(A))$ for every $A \subseteq \mathrm{supp}(\pi_y) \cap \mathrm{supp}(\pi_{s_i(y)})$.

Definition 4.4 is quite broad in that we make no assumptions on the behavior of the class-conditional densities outside of the overlapping regions. We now generalize Proposition 4.2.

**Theorem 4.5.** Let $\mathcal{X}$ consist of $N$ i.i.d. draws from any distribution $\pi$ satisfying Definition 4.4, and let $r \in \mathbb{R}$ be such that the sphere with radius $r$ in $\mathbb{R}^d$ has volume $k/(2MN)$. Then the result of Proposition 4.2 still holds for the set $\mathcal{S}_d$ of ERM interpolators for $\mathcal{X}$ which are $r$-regular over each overlapping region in $\mathrm{supp}(\pi_X)$.

To generalize Proposition 4.3, however, we need further restrictions on $\pi$. Mainly, we need to have significant spacing between non-overlapping classes (as in Definition 4.1), and we need to restrict the class-conditional densities such that mixings in $\mathcal{X}_{\mathrm{mix},d}$ are not too skewed towards a small subset of classes. The precise formulation of this assumption can be found in Appendix A.2.

**Theorem 4.6.** Let $\mathcal{X}$ consist of $N$ i.i.d. draws from any distribution $\pi$ satisfying Definition 4.4 and Assumption A.3, and let $p(k)$ be as in Proposition 4.3. Then the result of Proposition 4.3 still holds when considering $d$-Mixup interpolators $g$ for $\mathcal{X}$ where the mixing distribution $\mathcal{D}_{\lambda,d}$ is uniform over the $d$-dimensional probability simplex.

## 5 EXPERIMENTS

We now verify that the theoretical phenomenon of ERM calibration performance decreasing with distributional overlap also manifests in practice. For all of the experiments in this section, we train ResNeXt-50 models (Xie et al., 2016) due to the continued use of the ResNet (He et al., 2015) family of models as strong baselines (Wightman et al., 2021), and we report the mean performance over 5 random initializations with 1 standard deviation error bounds. For metrics, we report negative-log-likelihood (NLL), ECE with binning using 15 uniform bins (as was done by Guo et al. (2017)), and also top-class adaptive calibration error (ACE) (Nixon et al., 2020) which is ECE but computed using equal-weight bins. We also use confidence histograms and reliability diagrams to visualize calibration. The former corresponds to histograms of the predicted probabilities for the positive class (or top-class probabilities in the multi-class setting), while the latter correspond to binning the top-class probability predictions and plotting the average accuracy in each bin. These visualizations are generated using the calibration library of Küppers et al. (2020).

All models were trained for 200 epochs using Adam (Kingma & Ba, 2015) with the standard hyperparameters of $\beta_1 = 0.9$, $\beta_2 = 0.999$, a learning rate of $0.001$, and a batch size of 500 on a single A5000 GPU using PyTorch (Paszke et al., 2019). We did not use any additional training heuristics such as Dropout (Srivastava et al., 2014). In preliminary experiments, we found that minor changes to these hyperparameters did not affect our experimental results so long as the training horizon was sufficiently long (so as to achieve interpolation of training data). For each training dataset considered in this section, we set aside 10% of the data for calibration.

### 5.1 SYNTHETIC DATA

We first consider a synthetic data model which closely resembles Definition 4.1: binary classification on overlapping Gaussian data. Namely, we consider class 1 points to be drawn from $\mathcal{N}(0, \mathrm{Id}_{300})$ (i.e. 300-dimensional mean zero Gaussian), and class 2 to be drawn from $\mathcal{N}(\mu, \mathrm{Id}_{300})$. The probability of overlap between the two classes is controlled by the choice of the mean $\mu$ of class 2.

We consider $\mu \in \{0.25 * \mathbf{1}, 0.05 * \mathbf{1}, 0.01 * \mathbf{1}\}$ (where $\mathbf{1}$ here denotes the all-ones vector in $\mathbb{R}^{300}$) to cover a range of overlaps (0.05 corresponds to roughly $1/\sqrt{d}$ here). We sample 4000 training data points and 1000 test data points according to the aforementioned distribution, with class 1 and class 2 being chosen uniformly.

We compare the performance of ERM with temperature scaling (TS) to that of Mixup *without* temperature scaling (which means the set aside calibration data is not used). Our implementation of temperature scaling follows that of Guo et al. (2017). For Mixup, we consider the usual Mixup formulation with uniform mixing distribution as well as $d$-Mixup with $d = 2, 4$ (i.e. mixing 3 and 5

points respectively) with uniform mixing (although we do not enforce the technical conditions in Definition A.1 for $d$-Mixup). Results are reported in Table 2.

| Training Method | Metric | $\mu = 0.25$ | $\mu = 0.05$ | $\mu = 0.01$ |
|---|---|---|---|---|
| ERM + TS | NLL | 0.2593 ±0.0341 | 3.7331 ±0.4986 | 4.2977 ±0.5641 |
| | ECE | **0.0407** ±0.0026 | 0.4047 ±0.0173 | 0.4686 ±0.0059 |
| | ACE | 0.3114 ±0.0520 | 0.2755 ±0.0225 | 0.2972 ±0.0167 |
| Mixup | NLL | **0.1912** ±0.0111 | 0.7487 ±0.0356 | 0.8247 ±0.0400 |
| | ECE | 0.1161 ±0.0096 | 0.1481 ±0.0227 | 0.1945 ±0.0204 |
| | ACE | **0.1629** ±0.0192 | 0.1456 ±0.0218 | 0.2174 ±0.0416 |
| 2-Mixup | NLL | 0.2205 ±0.0338 | 0.7197 ±0.0150 | 0.7996 ±0.0331 |
| | ECE | 0.1490 ±0.0226 | 0.1169 ±0.0095 | 0.1842 ±0.0292 |
| | ACE | 0.1918 ±0.0282 | **0.1192** ±0.0175 | **0.2085** ±0.0324 |
| 4-Mixup | NLL | 0.2729 ±0.0299 | **0.6784** ±0.0079 | **0.7444** ±0.0254 |
| | ECE | 0.1979 ±0.0199 | **0.0549** ±0.0164 | **0.1018** ±0.0305 |
| | ACE | 0.1804 ±0.0103 | 0.1374 ±0.0370 | 0.2279 ±0.0408 |

Table 2: Mean NLL, ECE, and ACE on test data over 5 runs with 1 standard deviation error bounds on 2-class Gaussian data.

Our results are what we would expect from our theory – the NLL of the ERM model increases sharply as we increase the distributional overlap, while the Mixup models have comparatively small increases in NLL. Additionally, we find that the $d$-Mixup models perform better than the ERM and regular Mixup models as we increase the amount of overlap. Of note is the fact that this occurs even though this synthetic dataset consists of only 2 classes, suggesting that it is perhaps possible to improve our theoretical observations to a non-asymptotic setting with more precise analysis.

A reasonable concern with comparing model NLL is that it can correlate strongly with generalization performance. However, we can verify that in fact the underlying issue is one of calibration and not of generalization by comparing confidence histograms and reliability diagrams for the models being considered. Figure 2 compares the ERM + TS model to the 4-Mixup model; we see that the average positive class confidence for the 4-Mixup model concentrates around 0.5, while the ERM confidence is bimodal at 0 and 1 (i.e. extremely overconfident, even when making mistakes), and that the test accuracy of both models is roughly the same. Also, the observed underconfidence at lower probabilities for both models can be attributed to the lack of predictions at those probabilities. Additional plots for other models and levels of separation can be found in Appendix B.2.

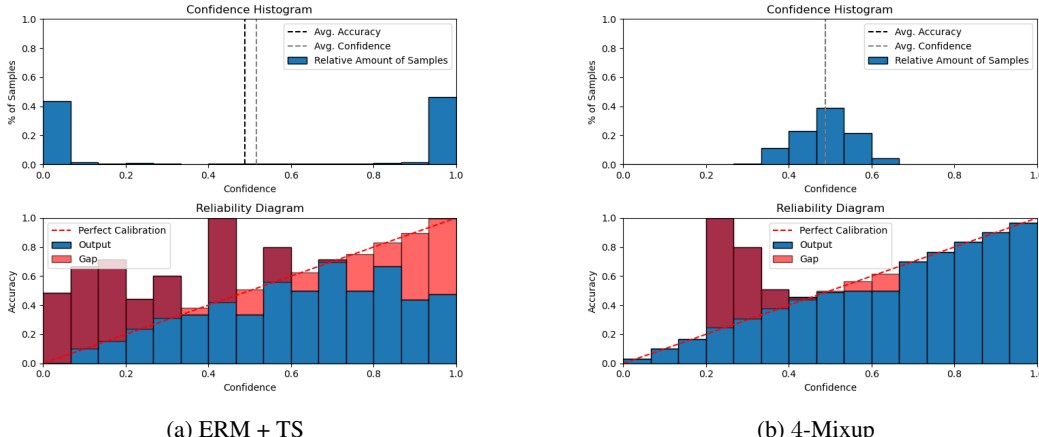

(a) ERM + TS

(b) 4-Mixup

Figure 2: Confidence histograms and reliability diagrams for ERM + TS and 4-Mixup models on the test Gaussian data with $\mu = 0.01 * \mathbf{1}$, using 15 bins. Overall accuracy on the test data, as well as average confidence, are reported as dashed lines on the histograms.

We observe a similar phenomenon to that of the NLL results when comparing model ECE, although we note that the ACE performance remains roughly the same across different overlap levels (this is outside the scope of our theory, since this is a byproduct of the binning procedure used for ACE). The

main takeaway from this set of experiments is that Mixup, and particularly the $d$-Mixup variants, have a *significant regularization effect on model confidence*. This regularization effect is more pronounced as we increase $d$ (as we would expect), but comes at a trade-off of additional computational overhead (computing convex combinations of $d + 1$ points per batch).

## 5.2 Image Classification Benchmarks

We can also verify that the phenomena observed in synthetic data translates to the more realistic benchmarks of CIFAR-10, CIFAR-100, and SVHN. To artificially introduce (more) overlaps in the datasets, we add label noise to the training data. In order to follow the style of Definition 4.4, our label noise procedure consists of randomly pairing up each class in the data with another distinct class, and then flipping the label of the original class points to be the label of the paired up class according to the desired level of label noise.

Results for ERM + TS and Mixup in terms of test NLL are shown in Table 3. Results in terms of ECE and ACE can be found in Appendix B.3; they follow similar trends to NLL. We additionally trained $d$-Mixup models on the same data, but we found that $d > 2$ led to underconfidence on these datasets, so we report just the results for Mixup. We also analyzed the relative improvement of TS over baseline ERM, as well as Mixup + TS; these results can be found in Appendix B.4.

| Dataset | Label Noise | ERM + TS (NLL) | Mixup (NLL) |
|---------|-------------|----------------|-------------|
| | 0% | 2.0109 $\pm$0.1750 | **0.8190** $\pm$0.0349 |
| CIFAR-10 | 25% | 3.2091 $\pm$0.2042 | **1.3102** $\pm$0.0311 |
| | 50% | 5.9533 $\pm$0.7209 | **1.7729** $\pm$0.0773 |
| | 0% | 5.4969 $\pm$0.8198 | **2.4608** $\pm$0.0482 |
| CIFAR-100 | 25% | 6.3767 $\pm$0.4089 | **2.9753** $\pm$0.1004 |
| | 50% | 8.0224 $\pm$0.6025 | **3.5443** $\pm$0.0367 |
| | 0% | 0.6919 $\pm$0.1178 | **0.3441** $\pm$0.0348 |
| SVHN | 25% | 1.6797 $\pm$0.2442 | **0.8546** $\pm$0.0358 |
| | 50% | 4.1807 $\pm$0.3642 | **1.5030** $\pm$0.0367 |

Table 3: Mean NLL over 5 runs with 1 standard deviation error bounds on image classification benchmark test data with varying levels of label noise.

We see in Table 3 the same behavior we observed in Section 5.1; the NLL for the ERM + TS models jumps quickly with increasing label noise while the Mixup NLL increases comparatively slowly. While in the cases of CIFAR-10 and CIFAR-100 some of this gap can be associated to improved test error of the Mixup models, we find for SVHN that ERM actually outperforms Mixup in terms of test error and still has massive gaps in terms of NLL. Furthermore, we show in Appendix B.3 that Mixup also dominates in terms of ECE and ACE across all datasets and virtually every label noise level. Finally, comparing confidence histograms and reliability diagrams in the manner of Section 5.1 shows the same behavior on the classification benchmarks: ERM + TS confidences cluster around 1 while Mixup confidences are more evenly distributed, contributing to improved calibration.

## 6 Conclusion

The key finding of our work is that when considering interpolating models on data distributions with overlaps, temperature-scaling-type methods alone can be insufficient for obtaining good calibration, both in theory and in practice. We show empirically that such interpolating models have very skewed confidence distributions (i.e. always predicting near 1), which re-affirms earlier empirical findings in the literature and suggests that temperature scaling is forced to push such models to be closer to random (as seen in Theorem 4.5). On the other hand, we also show that minimizing the Mixup-augmented loss instead of the standard empirical risk can lead to good calibration even when the data distribution has significant overlaps. Towards this end, we introduce $d$-Mixup, and show that mixing more than two points with Mixup can in fact be beneficial for calibration due to the added regularization, particularly in high noise/overlap settings. One clear direction suggested by our work is the idea of developing better *neighborhood constraints* around training points; we study the constraints introduced by Mixup, but we anticipate there are likely better alternatives for improving calibration that rely on non-linear constraints, and we leave this to future work.

ACKNOWLEDGEMENTS

Rong Ge and Muthu Chidambaram are supported by NSF Award DMS-2031849, CCF-1845171 (CAREER), CCF-1934964 (Tripods), and a Sloan Research Fellowship. Muthu would like to thank Kai Xu for helpful discussions during the early stages of this project.

ETHICS STATEMENT

Due to the largely theoretical nature of this work, we do not anticipate any misuses of our results or code. That being said, we do note that our work further emphasizes the calibration issues that can exist with present day models, which is an important reminder for those using these models for critical use cases.

REPRODUCIBILITY STATEMENT

All of the proofs and full assumptions for the theoretical results in this paper can be found in Appendix A. Code used to generate all of the plots in this paper can be found in the associated Github Repository: https://github.com/2014mchidamb/temp-scaling-limitations.

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

# A FULL PROOFS

Here we provide full proofs of all results in Section 4. For convenience, we first recall all of the relevant definitions and assumptions from Sections 2 and 4.

**Definition 3.1.** [ERM Interpolator] For a dataset $\mathcal{X}$, we say that a model $g$ is an ERM interpolator if for every $(x_i, y_i) \in \mathcal{X}$ there exists a universal constant $C_i$ such that:

$$\min_{s \neq y_i} g^{y_i}(x_i) - g^s(x_i) > \log k \quad \text{and} \quad \max_{r,s \neq y_i} g^s(x_i) - g^r(x_i) < C_i \tag{3.1}$$

**Definition 3.2.** [$\gamma$-Regular] For a point $(x_i, y_i) \in \mathcal{X}$, letting $L$ be a universal constant, we define:

$$\mathcal{B}_\gamma(x_i) = \{x \in \mathbb{R}^d : \|x_i - x\| \leq \gamma\} \tag{3.2}$$

$$\mathcal{G}_\gamma(x_i) = \{x \in \mathbb{R}^d : |g^{y_i}(x_i) - g^{y_i}(x)| \leq L\gamma\} \tag{3.3}$$

We say that a model $g$ is $\gamma$-regular over a set $U$ if there exists a class $y \in [k]$ and $\Theta(\pi_X(U)N)$ points $(x_i, y) \in \mathcal{X}$ with $x_i \in U$ such that $\pi_y(X \in \mathcal{G}_\gamma(x_i) \mid X \in \mathcal{B}_\gamma(x_i)) \geq 1 - O(1/k)$.

**Definition 3.4.** [$d$-Mixup Interpolator] For a dataset $\mathcal{X}$, we say that $g$ is a $d$-Mixup interpolator if $\phi^y(g(z)) = \phi^y(g^*(z)) \pm O(1/k)$ for almost every $z \in \mathcal{X}_{\text{mix},d}$ and $y \in [k]$, with $g^* \in \arginf_g J_{\text{mix},d}(g, \mathcal{X}, \mathcal{D}_{\lambda,d})$.

**Definition 4.1.** [$\alpha$-Overlapping Intervals] Let $\tau(y)$ denote the parity of a nonnegative integer $y$ and let $\beta_y = \lfloor (y-1)/2 \rfloor k + \alpha \tau(y-1)$ for $y \in [k]$, where $\alpha \in [0, 1]$ is a parameter of the distribution. Then we define $\pi(X, Y)$ to be the distribution on $\mathbb{R} \times [k]$ such that $\pi_Y$ is uniform over $[k]$ and $\pi(X \mid Y = y)$ is uniform over $[\beta_y, \beta_y + 1]$.

**Definition 4.4.** [General Data Distribution] Given a parameter $\alpha \in [0, 1]$, we define $\pi$ to be any distribution whose support is contained in $\mathbb{R}^d \times [k]$ satisfying the following constraints:

1. (Classes are roughly balanced) $\pi_Y(Y = y) = \Theta(1/k)$.

2. (Constant class overlaps) Letting $M$ denote a nonnegative integer constant, there exist $\Theta(k)$ classes $y$ for which there are classes $s_1(y), s_2(y), ..., s_m(y)$ for some $1 \leq m \leq M$ with $\pi_y(\text{supp}(\pi_y) \cap \text{supp}(\pi_{s_i(y)})) \geq 1 - \alpha$, and all other $s' \in [k]$ satisfy $\pi_X(\text{supp}(\pi_y) \cap \text{supp}(\pi_{s'})) = 0$.

3. (Overlap density is proportional to measure) $\pi_y(X \in A) = \Theta(\mu_d(A))$ and $\pi_{s_i(y)}(X \in A) = \Theta(\mu_d(A))$ for every $A \subseteq \text{supp}(\pi_y) \cap \text{supp}(\pi_{s_i(y)})$.

## A.1 PROOFS OF PROPOSITION 4.2 AND THEOREM 4.5

**Proposition 4.2.** Let $\mathcal{X}$ consist of $N$ i.i.d. draws from the distribution $\pi$ specified in Definition 4.1, with a parameter $\alpha$. Then with probability at least $1 - k \exp(-\Omega(N/k))$ over the randomness of $\mathcal{X}$, the set $\mathcal{S}$ of all models $g$ that are ERM interpolators for $\mathcal{X}$ and $k/(4N)$-regular over each overlapping region in $\text{supp}(\pi_X)$ is non-empty (in fact, uncountable). Furthermore, the predictive distribution $\hat{\pi}_T(Y \mid X) = \phi^Y(g_T(X))$ of the optimally temperature-scaled model $g_T$ for any $g \in \mathcal{S}$ satisfies:

$$\mathbb{E}_{X \sim \pi_X}[d_{\text{KL}}(\pi(Y \mid X), \hat{\pi}_T(Y \mid X))] \geq \Theta((1 - \alpha - 1/k) \log k) \tag{4.1}$$

Thus, for $\alpha = O(1)$, even with oracle temperature scaling every $g \in \mathcal{S}$ is asymptotically no better than random. In contrast, as the separation $\alpha \to 1$, the bound in Equation (4.1) becomes vacuous.

*Proof.* We begin by first verifying that the set $\mathcal{S}$ is large with high probability. Firstly, there are uncountably many strong ERM interpolators $g$, since the interpolator constraint is only on the finitely many points in $\mathcal{X}$ (so behavior is unconstrained elsewhere on the input space), and with probability 1 these points do not coincide.

In order to show that $k/(4N)$-regularity is feasible over each overlapping region, let us focus on a single class $y < k$ with $\tau(y) = 1$ (i.e. the class $y$ is odd-numbered). In this case, the overlapping region is $[\beta_y + \alpha, \beta_y + 1]$ (see Definition 4.1). By a Chernoff bound, there are $\Theta((1 - \alpha)N/k)$ class $y$ points in this region with probability $1 - \exp(-\Omega((1 - \alpha)N/k))$.

Now conditioning on each class $y$ point $x$ in the region, the probability that a class $y + 1$ point falls in a $k/(4N)$-neighborhood of $x$ is at most $1/(2N)$ (since the class-conditional density of the class $y + 1$ points is uniform over $[\beta_y + \alpha, \beta_y + 1 + \alpha]$, and the class prior is uniform over $[k]$). Thus, by a union bound the probability that $x$ has no class $y + 1$ neighbors is at least $1/2$. Now by another Chernoff bound, there are $\Theta((1 - \alpha)N/k)$ class $y$ points with no $k/(4N)$-neighborhood neighbors with probability at least $1 - \exp(-\Omega((1 - \alpha)N/k))$, which allows us to estbalish $k/(4N)$-regularity over $[\beta_y + \alpha, \beta_y + 1]$. Repeating this logic for each overlapping region (conditioning appropriately) and taking a union bound shows that the set $\mathcal{S}$ is large with probability $1 - k \exp(-\Omega((1 - \alpha)N/k))$.

With $k/(4N)$-regularity established, our approach is straightforward: we will lower bound the expected KL divergence by considering just the divergence over the regions consisting of the unions of the regularity neighborhoods, over which temperature scaling can at best bring the softmax outputs to uniform over $k$ (i.e. each output is $1/k$). However, the ground truth conditional distribution over the regularity neighborhoods is uniform over only two classes (since the neighborhoods are in a region of overlap between two classes).

As before, for simplicity we will focus on a single class since the argument that follows can be iterated for each subsequent class. To keep notation brief, let us define:

$$H_\pi(\hat{\pi} \mathbb{1}_A) = \mathbb{E}_{\pi(Y|X)}[-\mathbb{1}_A \log \hat{\pi}(Y \mid X)] \tag{A.1}$$

In other words, the cross-entropy restricted to a region $A$. Now consider a class $1 < y < k$ with $\tau(y) = 0$ (even parity); over its support $[\beta_y, \beta_y + 1]$ we observe that:

$$H_\pi(\pi \mathbb{1}_{[\beta_y, \beta_y+1]}) = -\frac{1}{k} \int_{\beta_y}^{\beta_y+1} \sum_{y \in [k]} \pi(Y = y \mid X = x) \log \pi(Y = y \mid X = x) \, dx$$

$$= \frac{\log 2}{k} \tag{A.2}$$

So the entropy over the support of class $y$ of the ground truth conditional distribution is a constant divided by $k$. We will now show that the cross-entropy over the same region for the temperature-scaled predictive distribution $\hat{\pi}_T(Y \mid X)$ is lower bounded by $\Theta((1 - \alpha) \log(k)/k)$.

To do so, we will constrain our attention as mentioned earlier to the entropy over just the regions specified by $k/(4N)$-regularity. Let $U$ denote the union of all of the $k/(4N)$-confidence neighborhoods around class $y - 1$ points in the region $[\beta_y, \beta_y + 1]$. Note that we are focusing on the neighborhoods around class $y - 1$ points (this corresponds to the odd-numbered class in our discussion above), because in these neighborhoods the class $y$ softmax output will be extremely small (due to the interpolation property applied to $g$ on the class $y - 1$ points). We have that:

$$H_\pi(\hat{\pi}_T \mathbb{1}_{[\beta_y, \beta_y+1]}) \geq -\frac{1}{k} \min_{T>0} \int_U \frac{1}{2} \log \phi^y(g(x)/T) \, dx \tag{A.3}$$

Now we can lower bound the integrand in Equation (A.3) at each point $(x_i, y - 1) \in \mathcal{X}$ with $x_i \in U$ as:

$$-\log \phi^y(g(x_i)/T) = -\log \frac{\exp(g^y(x_i)/T)}{\exp(g^{(y-1)}(x_i/T)) + \sum_{s \neq (y-1)} \exp(g^s(x_i)/T)}$$

$$\geq -\log \frac{\exp(g^y(x_i)/T)}{\exp(g^{(y-1)}(x_i/T)) + (k-1)\exp((g^y(x_i) - C_i)/T)} \tag{A.4}$$

Where $C_i$ above is from the ERM interpolation property of $g$. Applying $k/(4N)$-regularity we can translate the bound in Equation (A.4) to all $x \in U$ at the cost of introducing a $O(k/N)$ correction to $g^{(y-1)}(x_i/T)$ and a $O(1/k)$ correction to $\pi_X(U)$. The former error term will be irrelevant to the calculation (asymptotically), while the latter error term will show up in the final bound. Letting $\mathcal{X}_U$ denote all points $x_i \in U$ with $(x_i, y - 1) \in \mathcal{X}$ and recalling that $\mu_1(U)$ denotes the 1-dimensional Lebesgue measure of the set $U$, we then obtain:

$$H_\pi(\hat{\pi}_T \mathbb{1}_{[\beta_y, \beta_y+2]}) \geq$$

$$\Theta\left(-\frac{\mu_1(U) - O(1/k)}{k} \max_{x_i \in \mathcal{X}_U} \log \frac{\exp(g^y(x_i)/T)}{\exp(g^{(y-1)}(x_i/T)) + (k-1)\exp((g^y(x_i) - C_i)/T)}\right) \tag{A.5}$$

Now we consider two cases; $T < 1$ and $T \geq 1$. In the former case, since $\exp\big(g^{(y-1)}(x_i)\big) \geq k \exp(g^y(x_i))$, we have that the term inside the log is bounded above by $1/k$. In the latter case, since $C_i$ is a constant, the term inside the log is also bounded above by $1/k$. Thus we get:

$$H_\pi(\hat{\pi}_T \mathbb{1}_{[\beta_y, \beta_y+1]}) \geq \Theta\left(\frac{(\mu_1(U) - O(1/k))\log k}{k}\right) \tag{A.6}$$

Finally, we observe that $U$ consists of the union of $\Theta((1-\alpha)N/k)$ neighborhoods of size $\Theta(k/N)$, so $\mu_1(U) = \Theta(1-\alpha)$. Iterating this argument over each of the $\lfloor k/2 \rfloor$ overlapping regions, we obtain the desired result. $\qquad\square$

As mentioned, the proof of the more general result follows the proof of Proposition 4.2 extremely closely, and we thus recommend reading the above proof before proceeding to this one (as we skip some details).

**Theorem 4.5.** Let $\mathcal{X}$ consist of $N$ i.i.d. draws from any distribution $\pi$ satisfying Definition 4.4, and let $r \in \mathbb{R}$ be such that the sphere with radius $r$ in $\mathbb{R}^d$ has volume $k/(2MN)$. Then the result of Proposition 4.2 still holds for the set $\mathcal{S}_d$ of ERM interpolators for $\mathcal{X}$ which are $r$-regular over each overlapping region in $\text{supp}(\pi_X)$.

*Proof.* As in the proof of Proposition 4.2, there are uncountably many ERM interpolators for $\mathcal{X}$. Now for showing that there is a subset of such interpolators satisfying uniform $r$-confidence over overlapping regions with high probability, let us consider a class $y$ whose support overlaps with the supports of other classes $s_1(y), ..., s_m(y)$ (as defined in Definition 4.4).

Let $A \subset \text{supp}(\pi_y)$ denote the overlap between the support of $y$ and an arbitrary one of the aforementioned classes $s_i(y)$. As before, by a Chernoff bound, there are $\Omega((1-\alpha)N/k)$ class $y$ points falling in $A$ with probability at least $1 - \exp(-\Omega((1-\alpha)N/k))$, since by assumption $\pi_y(A) \geq 1 - \alpha$.

Conditioning on each class $y$ point $x$ in $A$, the probability that a point from classes $s_1(y), ..., s_m(y)$ falls in an $r$-neighborhood of $x$ is at most $m/(2MN)$. As before, this implies by a union bound and another Chernoff bound that there are $\Theta((1-\alpha)N/k)$ class $y$ points in the overlapping region with no $r$-neighborhood neighbors with probability at least $1 - \exp(-\Omega((1-\alpha)N/k))$. Repeating this logic and taking another union bound, it follows that the set $\mathcal{S}$ is large with probability at least $1 - k\exp(-\Omega((1-\alpha)N/k))$.

The rest of the proof carries over similarly from the proof of Proposition 4.2. Let $y$ be any class with overlaps, as considered above. Similarly, let $s$ be any class overlapping with $y$ whose region of overlap we denote as $A$, once again as above. Defining $H_\pi(\cdot)$ as in Equation (A.1), we see that:

$$H_\pi\left(\pi \mathbb{1}_{\text{supp}(\pi_s)}\right) = \Theta\left(\frac{1}{k}\right) \tag{A.7}$$

Now letting $U$ denote the union of all $r$-neighborhoods around class $y$ points in the overlapping region $A$, we also get:

$$H_\pi(\hat{\pi}_T \mathbb{1}_{\text{supp}(\pi_s)}) \geq -\Theta\left(\frac{1}{k} \min_{T>0} \int_U \log \phi^s(g(x)/T)\, dx\right) \tag{A.8}$$

As in the proof of Proposition 4.2, we can lower bound the integrand in Equation (A.8) at each point $(x_i, y) \in \mathcal{X}$ with $x_i \in U$ as:

$$-\log \phi^s(g(x_i)/T) \geq -\log \frac{\exp(g^s(x_i)/T)}{\exp(g^y(x_i/T)) + (k-1)\exp((g^s(x_i) - C_i)/T)} \tag{A.9}$$

Once again letting $\mathcal{X}_U$ denote all points $x_i \in U$ with $(x_i, y) \in \mathcal{X}$, we obtain from $r$-regularity:

$$H_\pi(\hat{\pi}_T \mathbb{1}_{\pi_s}) \geq \Theta\left(-\frac{\mu_d(U) - O(1/k)}{k} \max_{x_i \in \mathcal{X}_U} \log \frac{\exp(g^s(x_i)/T)}{\exp(g^y(x_i/T)) + (k-1)\exp((g^s(x_i) - C_i)/T)}\right) \tag{A.10}$$

Which, by identical logic to the proof of Proposition 4.2, gives:

$$H_\pi(\hat{\pi}_T \mathbb{1}_{\pi_s}) \geq \Theta\left(\frac{(\mu_d(U) - O(1/k)) \log k}{k}\right) \tag{A.11}$$

Observing again that $\mu_d(U) = \Omega(1 - \alpha)$ and applying this logic to all $\Theta(k)$ overlapping regions in $\mathrm{supp}(\pi_X)$, we obtain the desired result. $\qquad\square$

## A.2 MIXUP OPTIMALITY LEMMA AND PROOFS OF PROPOSITION 4.3 AND THEOREM 4.6

First, we provide a proper definition of the mixing set $\mathcal{M}_d(\mathcal{X})$, which was intuitively described in Section 3.2 of the main paper.

**Definition A.1.** Given a dataset $\mathcal{X}$, we define:

$$\mathcal{M}_d(\mathcal{X}) = \Big\{ \sigma \in [N]^{d+1} : x_{\sigma_i} - x_{\sigma_{d+1}} \neq 0 \quad \forall i \in [d]$$

$$\text{and} \quad \frac{\langle x_{\sigma_i} - x_{\sigma_{d+1}}, x_{\sigma_j} - x_{\sigma_{d+1}} \rangle}{\|x_{\sigma_i} - x_{\sigma_{d+1}}\| \|x_{\sigma_j} - x_{\sigma_{d+1}}\|} \leq \frac{1}{2d} \quad \forall i, j \in [d], \ i \neq j$$

$$\text{and} \quad \|x_{\sigma_i} - x_{\sigma_j}\| \leq C(\mathcal{X}) \quad \forall i, j \in [d+1] \Big\}$$

Where $C(\mathcal{X})$ is a possibly data-dependent constant.

We recall that the definition of $\mathcal{X}_{\mathrm{mix},d}$ is all $z$ such that $z = \sum_{j \in [d+1]} \lambda_j x_{\sigma_j}$ for some $\lambda \in \mathrm{supp}(\mathcal{D}_{\lambda,d})$ and $\sigma \in \mathcal{M}_d(\mathcal{X})$. Now we prove a formal version of Lemma 3.3, which we will rely on in the proofs of Proposition 4.3 and Theorem 4.6.

**Lemma A.2.** Let $f_\lambda$ denote the density of $\mathcal{D}_{\lambda,d}$ and define:

$$L_\sigma = \begin{bmatrix} x_{\sigma_1} - x_{\sigma_{d+1}} & x_{\sigma_2} - x_{\sigma_{d+1}} & \cdots & x_{\sigma_d} - x_{\sigma_{d+1}} \end{bmatrix} \tag{A.12}$$

$$\Lambda_\sigma(z) = \begin{bmatrix} L_\sigma^{-1}(z - x_{\sigma_{d+1}}) & 1 - \sum_{j \in [d]} L_\sigma^{-1}(z - x_{\sigma_{d+1}})_j \end{bmatrix} \tag{A.13}$$

$$\xi_y(z) = \sum_{\sigma \in \mathcal{M}_d(\mathcal{X})} \mathbb{1}_{z \in \mathrm{conv}(\{x_{\sigma_i}\}_{i=1}^{d+1})} |\det(L_\sigma^{-1})| f_\lambda(\Lambda_\sigma(z)) \sum_{j: \, y_{\sigma_j} = y} \Lambda_\sigma(z)_j \tag{A.14}$$

Then any $g^* \in \mathrm{arginf}_g J_{\mathrm{mix},d}(g, \mathcal{X}, \mathcal{D}_{\lambda,d})$ satisfies $\phi^y(g^*(z)) = \xi_y(z)/\sum_{s \in [k]} \xi_s(z)$ for almost every $z \in \mathcal{X}_{\mathrm{mix},d}$.

*Proof.* First, writing out the expectation in Equation (3.5), we have:

$$J_{\mathrm{mix},d}(g, \mathcal{X}, \mathcal{D}_{\lambda,d}) = -\frac{1}{|\mathcal{M}_d(\mathcal{X})|} \sum_{\sigma \in \mathcal{M}_d(\mathcal{X})} \int_{\Delta^d} \sum_{i \in [d+1]} \lambda_i \log \phi^{y_{\sigma_i}} \left( g\left( \sum_{j \in [d+1]} \lambda_j x_{\sigma_j} \right) \right) f(\lambda) \, d\lambda \tag{A.15}$$

Now we make the substitution $z = \sum_{j \in [d+1]} \lambda_j x_{\sigma_j}$. By the definition of $\mathcal{M}_d(\mathcal{X})$, the transformation $L_\sigma$ in Equation (A.12) is invertible, so $\Lambda_\sigma(z)$ from Equation (A.13) is well-defined. Now applying Fubini's Theorem, we may write:

$$\ell(\sigma, z) = -|\det(L_\sigma^{-1})| f_\lambda(\Lambda_\sigma(z)) \sum_{j \in [d+1]} \Lambda_\sigma(z)_j \log \phi^{y_{\sigma_j}}(g(z)) \tag{A.16}$$

$$J_{\mathrm{mix},d}(g, \mathcal{X}, \mathcal{D}_{\lambda,d}) = \frac{1}{|\mathcal{M}_d(\mathcal{X})|} \int_{\mathcal{X}_{\mathrm{mix},d}} \sum_{\sigma \in \mathcal{M}_d(\mathcal{X})} \mathbb{1}_{z \in \mathrm{conv}(\{x_{\sigma_i}\}_{i=1}^{d+1})} \ell(\sigma, z) \, dz \tag{A.17}$$

We observe that $\sum_{\sigma \in \mathcal{M}_d(\mathcal{X})} \mathbb{1}_{z \in \text{conv}(\{x_{\sigma_i}\}_{i=1}^{d+1})} \ell(\sigma, z)$ is strictly convex as a function of $\phi(g(z))$. Now we can optimize the aforementioned term as a function of $\phi(g(z))$ under the constraint that $\sum_{y \in [k]} \phi^y(g(z)) = 1$. This is straightforward to do with Lagrange multipliers, and the solution is of the form $\phi^y(g(z)) = \xi_y(z)/\sum_{s \in [k]} \xi_s(z)$ where $\xi_y(z)$ is defined as in Equation (A.14), so the result follows. $\qquad\square$

With Lemma A.2, we can prove Proposition 4.3.

**Proposition 4.3.** Let $\mathcal{X}$ be as in Proposition 4.2 and $p(k)$ denote a polynomial in $k$ of degree at least one. Then taking $\mathcal{D}_{\lambda,1}$ to be uniform, *every* 1-Mixup interpolator $g$ for $\mathcal{X}$ with the property that $\phi^y(g(x)) \leq 1 - \Omega(1/p(k))$ for every $x \in \text{supp}(\pi_X) \setminus \mathcal{X}_{\text{mix},1}$ and $y \in [k]$ satisfies with probability at least $1 - k^3 \exp(-\Omega(N/k^3))$:

$$\mathbb{E}_{X \sim \pi_X}[d_{\text{KL}}(\pi(Y \mid X), \hat{\pi}(Y \mid X))] \leq \Theta(1) \tag{4.2}$$

Note that this result is independent of the separation parameter $\alpha$.

*Proof.* We first partition $\text{supp}(\pi_X)$ into subregions each of which have measure $\Theta(1/k^2)$ (the reason for this choice will be made clear shortly), so that there are $\Theta(k^3)$ total subregions. By a Chernoff bound, there are $\Theta(N/k^3)$ points in each of these subregions with probability at least $1 - \exp(-\Omega(N/k^3))$. Thus, by a union bound, the probability that every subregion contains $\Theta(N/k^3)$ points is at least $1 - k^3 \exp(-\Omega(N/k^3))$. Consequently, with the same probability, $\pi_X(\text{supp}(\pi_X) \setminus \mathcal{X}_{\text{mix},d}) = O(1/k^2)$. To see this, observe that for $z$ to be in $\mathcal{X}_{\text{mix},d}$, we need only have one point to the left of $z$ and one point to the right of $z$, both within a constant distance of one another (note that the other conditions in $\mathcal{M}_d(\mathcal{X})$ are vacuously satisfied since we are in dimension 1). By our high probability arguments, there is at most some $O(1/k^2)$ Lebesgue measure region in each pair of overlapping class intervals with no points in $\mathcal{X}$, from which it follows that $\pi_X(\text{supp}(\pi_X) \setminus \mathcal{X}_{\text{mix},d}) = O(1/k^2)$.

Now for $y \in [k]$, consider $z \in \text{supp}(\pi_y) \cap \mathcal{X}_{\text{mix},d}$. We will show that every 1-Mixup interpolator $g$ for $\mathcal{X}$ with $\mathcal{D}_{\lambda,1}$ being uniform satisfies $\phi^y(g(z)) \in [\Theta(1), 1]$ when $z$ falls only in the support of $y$ and $\phi^y(g(z)) \in [\Theta(1), 1 - \Theta(1)]$ in the overlapping region, which will be sufficient for showing the desired result.

We first handle the non-overlapping region of $\text{supp}(\pi_y)$. Here we need to consider the effect of the separation parameter $\alpha$; if $\alpha = O(1/k)$ (i.e. the classes have very high overlap), the behavior in this region will be asymptotically irrelevant to the final KL-divergence calculation (since it will be a $O(\log(k)/k)$ term). On the other hand, if $\alpha = \Omega(1/k)$, then our partitioning of the input space was fine enough that we may claim from our high probability arguments that there are $\Theta(\alpha N/k)$ class $y$ points in some subregion $[\beta_y, \beta_y + \Theta(\alpha)]$. Similar logic also applies when considering the overlapping region, but this time the aforementioned logic is applied to $1 - \alpha$. Thus, in what follows, we will assume we are in the regime of $\alpha \in [\Omega(1/k), 1 - \Omega(1/k)]$.

Suppose without loss of generality that $\tau(y) = 1$ and that $y < k$ (i.e. $y$ is an odd-numbered class), and consider $z \in [\beta_y, \beta_y + \alpha] \cap \mathcal{X}_{\text{mix},d}$ (this is, as mentioned, the non-overlapping part of $\text{supp}(\pi_y)$). Clearly $z$ can only be obtained from $\sigma \in \mathcal{M}_d(\mathcal{X})$ with $y_{\sigma_1} = y$ or $y_{\sigma_2} = y$, since the other class supports are spaced $k$ away and $z \in \text{supp}(\pi_y) \setminus \text{supp}(\pi_{y+1})$ (i.e. we have to mix with class $y$). This implies that for $g^* \in \text{arginf}_g J_{\text{mix},1}(g, \mathcal{X}, \mathcal{D}_{\lambda,1})$, we have $\xi_s(z) = 0$ for $s \neq y$, $y + 1$ (where $\xi_y$ is as in Equation (A.14)). Now we consider two sub-cases: $z \in [\beta_y, \beta_y + \alpha/2)$ and $z \in [\beta_y + \alpha/2, \beta_y + \alpha)$.

In the first sub-case, if $y_{\sigma_j} = y$ then $\lambda_j \geq 1/2$ since $z = \lambda_1 x_{\sigma_1} + (1 - \lambda_1)x_{\sigma_2}$ must necessarily fall closer to whichever class $y$ point produced it, which implies that $\xi_y(z) \geq \xi_{y+1}(z)$.

For the latter sub-case, this is no longer true, since some points in $[\beta_y + \alpha/2, \beta_y + \alpha)$ may fall closer to class $y + 1$. However, by our initial arguments there are $\Theta(\alpha N/k)$ class $y$ points in $[\beta_y, \beta_y + \alpha/2)$, which implies there are $\Theta(\alpha N^2/k^2)$ choices of $\sigma \in \mathcal{M}_d(\mathcal{X})$ which can produce $z \in [\beta_y + \alpha/2, \beta_y + \alpha)$ with a constant weight associated with class $y$. Since there are $O(\alpha N^2/k^2)$ total choices of $\sigma \in \mathcal{M}_d(\mathcal{X})$ that can produce $z \in [\beta_y + \alpha/2, \beta_y + \alpha)$ (since we need to mix with a point to the left of $z$), we get that $\xi_y(z) = \Omega(\xi_{y+1}(z))$ in this case. Now from Definition 3.4, it immediately follows that every 1-Mixup interpolator $g$ satisfies $\phi^y(g(z)) \in [\Theta(1), 1]$ for $z \in [\beta_y, \beta_y + \alpha)$.

Let us now consider $z \in [\beta_y + \alpha, \beta_y + 1] \cap \mathcal{X}_{\mathrm{mix},d}$ (the overlapping part of $\mathrm{supp}(\pi_y)$). We can apply similar logic to that used in handling the sub-case of $z \in [\beta_y + \alpha/2, \beta_y + \alpha)$ considered above; namely, for every $z \in [\beta_y + \alpha, \beta_y + 1]$, there are $\Theta((z - \beta_y)(\beta_y + 1 + \alpha - z)N^2/k^2)$ choices of $\sigma \in \mathcal{M}_d(\mathcal{X})$ which can produce $z$ with a constant weight associated with class $y$ (consider our partitioning of the space), and so as before we get that $\xi_y(z) = \Omega(\xi_{y+1}(z))$. However, there are also the same order of choices of $\sigma \in \mathcal{M}_d(\mathcal{X})$ which can produce $z$ with a constant weight associated with class $y + 1$, so we also get $\xi_{y+1}(z) = \Omega(\xi_y(z))$. Thus, it follows that $g$ satisfies $\phi^y(g(z)) \in [\Theta(1), 1 - \Theta(1)]$ for $z \in [\beta_y + \alpha, \beta_y + 1] \cap \mathcal{X}_{\mathrm{mix},d}$.

Based on the above, we see that $H_\pi(\hat{\pi}\mathbb{1}_{\mathcal{X}_{\mathrm{mix},1}}) = \Theta(1)$ (where we recall that $H_\pi$ is defined as in Equation (A.1)). Furthermore, since $\phi^y(g(x)) \leq 1 - \Omega(1/\mathrm{poly}(k))$ on $\mathrm{supp}(\pi_X) \setminus \mathcal{X}_{\mathrm{mix},1}$, we have $H_\pi(\hat{\pi}\mathbb{1}_{\mathrm{supp}(\pi_X) \setminus \mathcal{X}_{\mathrm{mix},1}}) = O(\log(k)/k)$, from which the overall result follows. $\qquad\square$

Before proving Theorem 4.6, we need to formally state the necessary assumptions alluded to in the main text. Once again, these generalize the core facets of the simple data distribution from Definition 4.1.

**Assumption A.3.** In what follows, $\mathcal{X}$ is understood to be $N$ i.i.d. draws from $\pi$ and $\mathcal{X}_{\mathrm{mix},d}$ is obtained from $\mathcal{X}$ as defined above and in Section 3.2. With this in mind, we restrict the class of distributions $\pi$ from Definition 4.4 such that:

1. (Non-overlapping classes are separated) For $y, s \in [k]$, if $\pi_X(\mathrm{supp}(\pi_y) \cap \mathrm{supp}(\pi_s)) = 0$, then $d(\mathrm{supp}(\pi_y), \mathrm{supp}(\pi_s)) = \omega(1)$. Additionally, $\mathrm{supp}(\pi_y) = \Theta(1)$ for all $y \in [k]$.

2. (Mixed points cover support) With probability at least $1 - k^3 \exp(-\Omega(N/k^3))$, $\pi_X(\mathrm{supp}(\pi_X) \setminus \mathcal{X}_{\mathrm{mix},d}) = O(1/k)$.

3. (Mixed points are balanced across overlapping classes) Let us define for $y \in [k]$ and $z \in \mathbb{R}^d$:

$$\mathcal{M}_d^y(\mathcal{X}, z) = \left\{ \sigma \in \mathcal{M}_d(\mathcal{X}) : \exists \lambda \in \mathrm{supp}(\mathcal{D}_{\lambda,d}) \,\middle|\, z = \sum_{j \in [d+1]} \lambda_j x_{\sigma_j} \right.$$

$$\left. \text{and} \quad \sum_{j:y_{\sigma_j}=y} \lambda_j = \Theta(1) \right\} \tag{A.18}$$

Then with probability at least $1 - k^3 \exp(-\Omega(N/k^3))$, for every $y \in [k]$ and $z \in \mathcal{X}_{\mathrm{mix},d} \cap \mathrm{supp}(\pi_y)$ we have that $|\mathcal{M}_d^y(\mathcal{X}, z)| = \Omega(f(z, \alpha)N^{d+1}/k^{d+1})$ for some function $f$, or we have for every $\sigma \in \mathcal{M}_d(\mathcal{X})$ such that $z = \sum_{j \in [d+1]} \lambda_j x_{\sigma_j}$ with $\lambda \in \mathrm{supp}(\mathcal{D}_{\lambda,d})$:

$$\sum_{j:y_{\sigma_j}=y} \lambda_j \geq \sum_{i:y_{\sigma_i}\neq y} \lambda_i \tag{A.19}$$

The third part of Assumption A.3 appears complicated, but it just generalizes one of the key ideas in the proof of Proposition 4.2: either our mixed points always fall closer to one class (i.e. they are far away from the overlapping region), or there are a lot of possible mixings that can produce the mixed point. Ideally we would have structured Assumption A.3 to be purely in terms of the shape of $\pi$ and not in terms of the high probability behavior of the mixed points $\mathcal{X}_{\mathrm{mix},d}$, but various edge cases in high dimensions led us to structuring the assumption this way.

With this assumption, it is straightforward to prove the generalization of Proposition 4.2.

**Theorem 4.6.** Let $\mathcal{X}$ consist of $N$ i.i.d. draws from any distribution $\pi$ satisfying Definition 4.4 and Assumption A.3, and let $p(k)$ be as in Proposition 4.3. Then the result of Proposition 4.3 still holds when considering $d$-Mixup interpolators $g$ for $\mathcal{X}$ where the mixing distribution $\mathcal{D}_{\lambda,d}$ is uniform over the $d$-dimensional probability simplex.

*Proof.* Assumption A.3 allows us to largely lift the proof of Proposition 4.2 to higher dimensions. Indeed, from the second part of Assumption A.3 we already begin with the fact that $\pi_X(\mathrm{supp}(\pi_X) \setminus \mathcal{X}_{\mathrm{mix},d}) = O(1/k)$ with probability at least $1 - k^3 \exp(-\Omega(N/k^3))$.

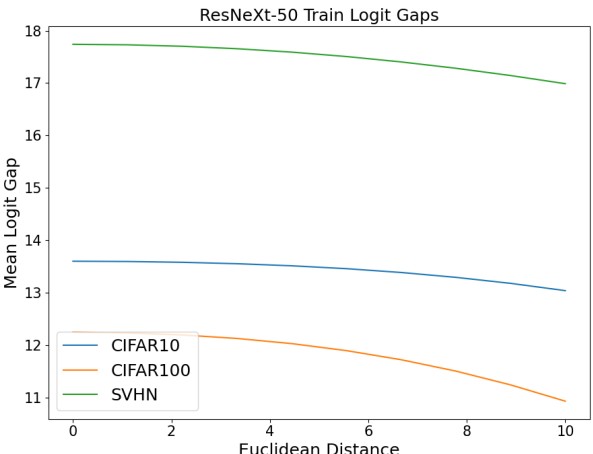

Figure 3: Mean logit gap change as a function of distance away from the original training point.

Now as before, for $y \in [k]$, consider $z \in \text{supp}(\pi_y) \cap \mathcal{X}_{\text{mix},d}$. By the first part of Assumption A.3 and Definition A.1, $z$ can only be obtained by mixing points from class $y$ and/or its overlapping neighbors $s_1(y), ..., s_m(y)$.

If $z \notin \text{supp}(\pi_{s_i(y)})$ for every $s_i(y)$ (i.e. it is not in the overlapping region), then by the third part of Assumption A.3 it immediately follows that $\phi^y(g(z)) \in [\Theta(1), 1]$ (note that unlike in dimension 1 the expression for $\xi_y$ in Lemma A.2 is more complicated now, but due to assuming a uniform mixing density and near orthogonality of mixed points in Definition A.1 we can simplify to essentially expressions like in the 1-d case). On the other hand, if $z = \sum_{j \in [d+1]} \lambda_j x_{\sigma_j} \in \text{supp}(\pi_{s_i(y)})$ for some $s_i(y)$, then we cannot simultaneously satisfy:

$$\sum_{j:y_{\sigma_j}=y} \lambda_j \geq \sum_{r:y_{\sigma_r}\neq y} \lambda_r \quad \text{and} \quad \sum_{j:y_{\sigma_j}=s_i(y)} \lambda_j \geq \sum_{r:y_{\sigma_r}\neq s_i(y)} \lambda_r \tag{A.20}$$

This implies that the other condition in the third part of Assumption A.3 is satisfied for both $y$ and $s_i(y)$; i.e. $|\mathcal{M}_d^y(\mathcal{X}, z)| = \Omega(f(z, \alpha)N^{d+1}/k^{d+1})$ (see Equation (A.18)). This again implies that $\phi^y(g(z)) \in [\Theta(1), 1 - \Theta(1)]$. As in the final part of the proof of Proposition 4.3, from this we get $H_\pi(\hat{\pi}\mathbb{1}_{\mathcal{X}_{\text{mix},d}}) = \Theta(1)$ and we are done from the fact that $\phi^y(g(x)) \leq 1 - \Omega(1/\text{poly}(k))$ on $\text{supp}(\pi_X) \setminus \mathcal{X}_{\text{mix},d}$.  □

# B  ADDITIONAL EXPERIMENT PLOTS

## B.1  MODEL LOGIT BEHAVIOR

To verify that the locally Lipschitz-like model behavior of Definition 3.2 is reasonable in our setting, we consider the following experimental setup. For every training point in each dataset we consider, we uniformly sample 500 points (due to compute constraints) from the surfaces of hyperspheres centered at the original training point with radii ranging from 1 to 10. The cutoff radius of 10 was chosen as that was roughly the minimum distance to a neighboring training point of a different class across the considered datasets.

For all sampled points, we compute the trained model (following the training setup of Section 5) logit gap (to account for translation invariance) between the logit for the correct class (or the original point) and the largest logit belonging to a different class. We then compute the mean over all logit gaps at each of the different radii considered, and plot how the mean logit gap changes as we move farther away from the original training points. Results are shown in Figure 3.

As can be seen in Figure 3, the mean logit gaps for each dataset barely move over this radius, suggesting that assuming Lipschitz-like behavior of the logits with high probability should be reasonable in small neighborhoods. However, we acknowledge that to truly make this claim one

would need to prove guarantees about the sample complexity needed to get appropriate coverage in each neighborhood. Our goal with the experiments in this section has only been to provide an initial attempt at such justification; we think that exploring how model logit/confidence behavior changes as one moves away from the training data contains many interesting challenges for future work.

## B.2 OMITTED PLOTS FOR SYNTHETIC DATA

Figures 4 to 6 display the confidence histograms and reliability diagrams for ERM + TS as well as all of the Mixup models (Mixup, 2-Mixup, and 4-Mixup) for the different settings of Gaussian test data in Section 5.1. As was originally seen in Table 2, the calibration performance for ERM markedly declines as we move from $\mu = 0.25$ to $\mu = 0.05$.

The reason for this is visible in the confidence diagrams: at the $\mu = 0.25$ level, the ERM confidences are still very bimodal, but the accuracy in the top bin is close to perfect. On the other hand, when moving to $\mu = 0.05$, the accuracy in the top bin drops significantly while the accuracy in the bottom bin increases, which corresponds to the drop in calibration performance since nearly all predictions fall in these two bins.

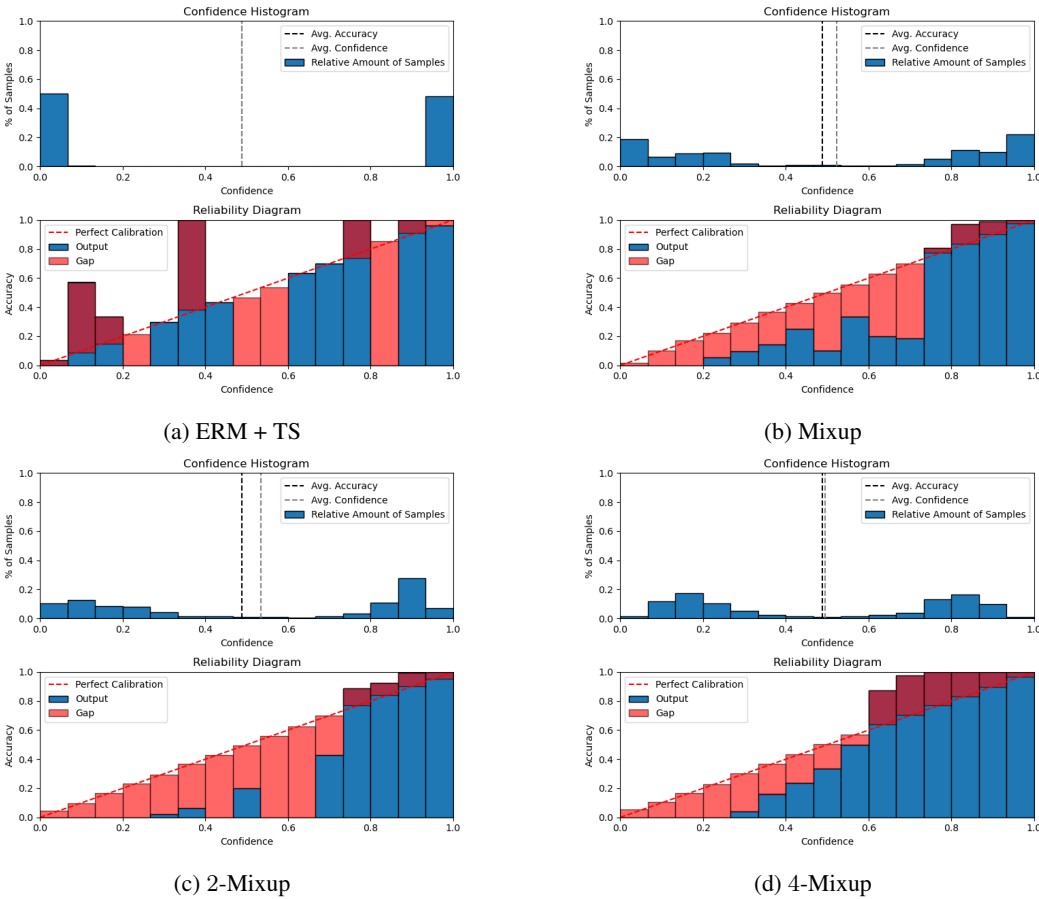

Figure 4: Confidence histograms and reliability diagrams for ERM + TS and Mixup models on the test Gaussian data with $\mu = 0.25 * \mathbf{1}$, using 15 bins.

## B.3 OMITTED PLOTS AND TABLES FOR IMAGE CLASSIFICATION BENCHMARKS

Tables 4 and 5 correspond to the ECE and ACE versions of 3; we find as mentioned in the main text that Mixup dominates in ECE and ACE as well. Unlike in the case of Gaussian data, here we see that both the ECE and ACE monotonically increase with label noise for the ERM + TS models (the only exception being SVHN at the 25% label noise level, for which the mean ACE decreases but remains

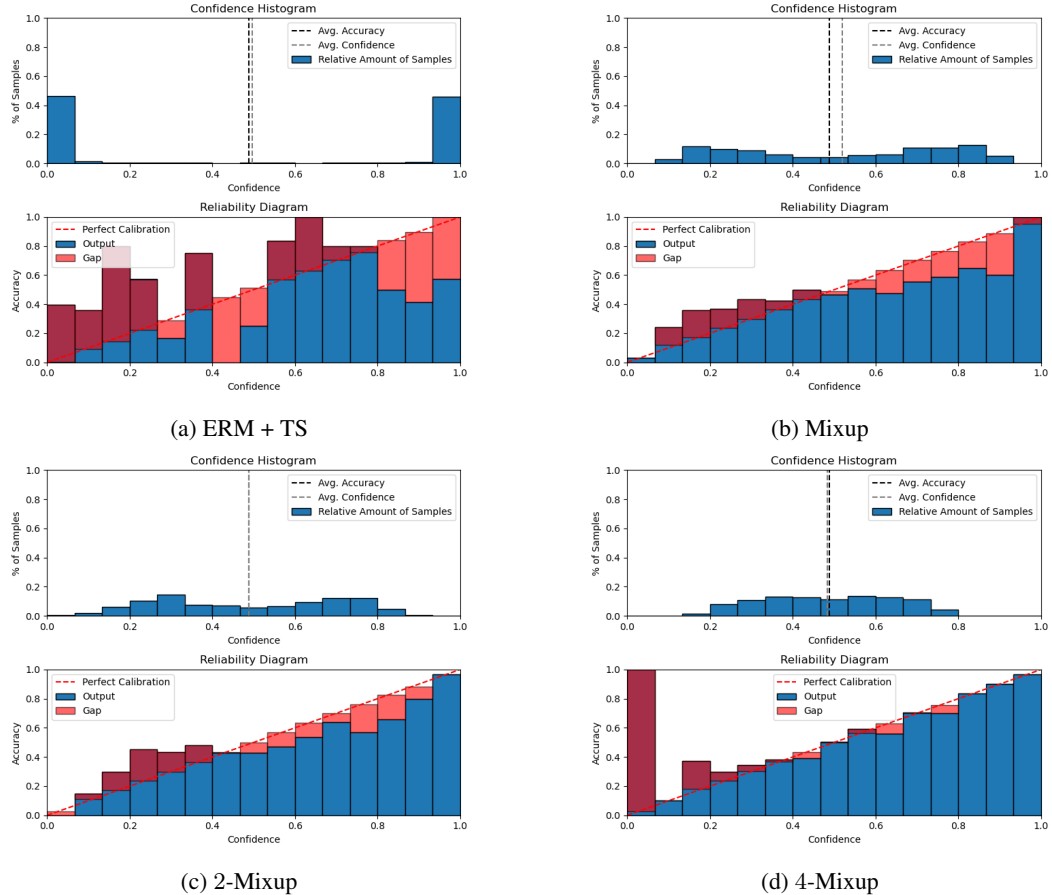

Figure 5: Confidence histograms and reliability diagrams for ERM + TS and Mixup models on the test Gaussian data with $\mu = 0.05 * \mathbf{1}$, using 15 bins.

within 1 standard deviation of the ACE for the 0% noise level). For Mixup, we find that there are a few additional instances in which mean ECE and ACE seemingly improve with increasing label noise (CIFAR-100 and SVHN), but again in these instances the performances are roughly within the one standard deviation error bounds of the lower noise levels. Furthermore, we also remark that is is possible for NLL to get worse (as seen in Table 3) while ECE/ACE improve. As an example, one can consider a uniformly random classifier - when classes are roughly balanced, this classifier has near-zero ECE despite having poor NLL performance.

Figures 7 to 9 contain the confidence histograms and reliability diagrams for the ERM + TS and Mixup models across CIFAR-10, CIFAR-100, and SVHN. Unlike in the case of Gaussian test data, we find non-trivial gaps in test accuracy between the ERM and Mixup models on CIFAR-10 and CIFAR-100, which certainly contributes to some of the gap in reported negative log-likelihood. However, we can see in Figure 9 that ERM actually outperforms Mixup in terms of test accuracy in this case (across different label noise settings), and still suffers a massive gap in NLL. Additionally, the confidence histograms are telling, as we once again find that the ERM + TS predictions cluster heavily in the largest probability bin, which is again the cause for poor calibration performance.

## B.4 ANALYSIS OF THE IMPACT OF TEMPERATURE SCALING

Tables 6 to 8 show the relative performance gains in terms of NLL, ECE, and ACE respectively due to temperature scaling for both ERM and Mixup on CIFAR-10, CIFAR-100, and SVHN. In the interest of space, we report the NLL results rounded to the nearest hundredth, and the ECE/ACE results rounded to the nearest tenth after multiplying by a factor of 100.

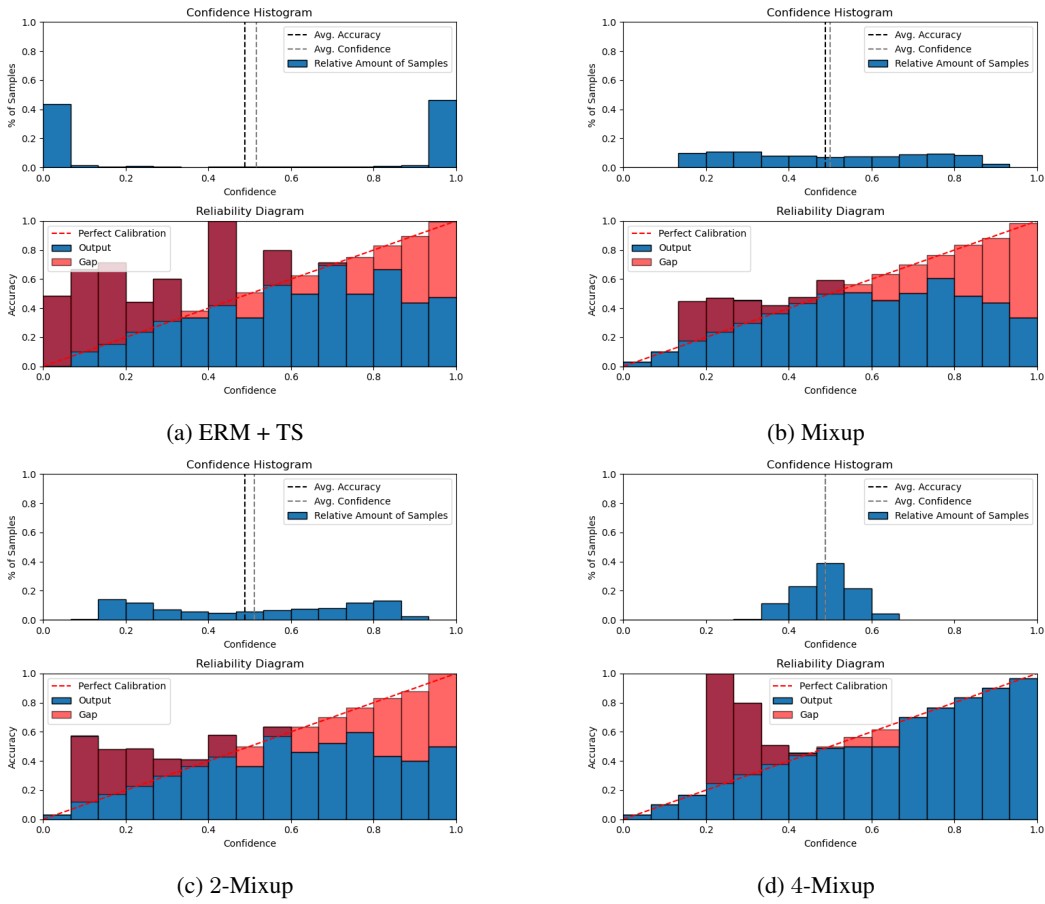

Figure 6: Confidence histograms and reliability diagrams for ERM + TS and Mixup models on the test Gaussian data with $\mu = 0.01 * \mathbf{1}$, using 15 bins.

| Dataset | Label Noise | ERM + TS (ECE) | Mixup (ECE) |
|---|---|---|---|
| CIFAR-10 | 0% | 0.2259 ±0.0057 | **0.0709** ±0.0103 |
| | 25% | 0.3540 ±0.0194 | **0.1055** ±0.0200 |
| | 50% | 0.5540 ±0.0111 | **0.1760** ±0.0569 |
| CIFAR-100 | 0% | 0.4370 ±0.0413 | **0.0811** ±0.0175 |
| | 25% | 0.4890 ±0.0164 | **0.0634** ±0.0159 |
| | 50% | 0.5846 ±0.0162 | **0.0649** ±0.0251 |
| SVHN | 0% | **0.0696** ±0.0051 | 0.0918 ±0.0083 |
| | 25% | 0.2070 ±0.0227 | **0.0892** ±0.0330 |
| | 50% | 0.4799 ±0.0175 | **0.1314** ±0.0255 |

Table 4: Mean ECE over 5 runs with 1 standard deviation error bounds on image classification benchmark test data with varying levels of label noise.

The main observations we make are that temperature scaling can make a significant difference for the ERM models, while it makes very little difference for the Mixup models. Furthermore, as one would expect, the greatest benefit is seen with respect to NLL, which is exactly the objective on which the temperature scaling parameter $T$ is fit. For ECE and ACE, we find that temperature scaling does not make a notable difference in our setup, as in most cases the post-TS performance is not outside of the one standard deviation error bounds of the pre-TS performance, although for ERM the mean ECE/ACE is almost always improved by temperature scaling.

| Dataset | Label Noise | ERM + TS (ACE) | Mixup (ACE) |
|---------|-------------|----------------|-------------|
| CIFAR-10 | 0% | 0.2562 ±0.0188 | **0.1012** ±0.0132 |
|  | 25% | 0.2909 ±0.0108 | **0.1216** ±0.0327 |
|  | 50% | 0.3691 ±0.0114 | **0.2060** ±0.0471 |
| CIFAR-100 | 0% | 0.3702 ±0.0250 | **0.1004** ±0.0197 |
|  | 25% | 0.3992 ±0.0193 | **0.0986** ±0.0245 |
|  | 50% | 0.4608 ±0.0218 | **0.1274** ±0.0550 |
| SVHN | 0% | 0.2436 ±0.0409 | **0.0643** ±0.0091 |
|  | 25% | 0.2049 ±0.0189 | **0.0937** ±0.0139 |
|  | 50% | 0.2851 ±0.0108 | **0.1880** ±0.0277 |

Table 5: Mean ACE over 5 runs with 1 standard deviation error bounds on image classification benchmark test data with varying levels of label noise.

| Dataset | Label Noise | ERM | ERM + TS | Mixup | Mixup + TS |
|---------|-------------|-----|----------|-------|-----------|
| CIFAR-10 | 0% | 2.21 ±0.20 | 2.01 ±0.18 | 0.82 ±0.04 | **0.82** ±0.04 |
|  | 25% | 3.53 ±0.23 | 3.21 ±0.20 | 1.31 ±0.03 | **1.30** ±0.03 |
|  | 50% | 6.57 ±0.81 | 5.95 ±0.72 | 1.77 ±0.08 | **1.76** ±0.07 |
| CIFAR-100 | 0% | 6.02 ±0.92 | 5.50 ±0.82 | 2.46 ±0.05 | **2.45** ±0.05 |
|  | 25% | 6.97 ±0.45 | 6.38 ±0.41 | 2.98 ±0.10 | **2.98** ±0.10 |
|  | 50% | 8.78 ±0.67 | 8.02 ±0.60 | 3.54 ±0.04 | **3.54** ±0.04 |
| SVHN | 0% | 0.76 ±0.13 | 0.69 ±0.12 | 0.34 ±0.04 | **0.33** ±0.04 |
|  | 25% | 1.84 ±0.27 | 1.68 ±0.24 | 0.86 ±0.04 | **0.86** ±0.04 |
|  | 50% | 4.61 ±0.41 | 4.18 ±0.36 | 1.50 ±0.04 | **1.50** ±0.04 |

Table 6: Mean NLL (rounded to the nearest hundredth) over 5 runs with 1 standard deviation error bounds on image classification benchmarks for TS and non-TS ERM and Mixup.

This is perhaps not super surprising in light of the reliability diagram and confidence histogram visualizations of Figures 7 to 9. For ERM, we would need a large choice of $T$ to make a substantial difference in the ECE/ACE performance due to the gap between the top-class logit/probability and the other classes, but this is of course not optimal when optimizing $T$ using the NLL objective. For Mixup, the pre-TS probabilities are already much less spiky (leading to good ECE/ACE because classes are roughly balanced in the datasets we consider), and they change negligibly after temperature scaling. We also note that while there are instances in which the post-TS Mixup ECE/ACE performance is worse than the pre-TS performance, in these cases the change in ECE/ACE is virtually nothing, and falls well within the one standard deviation error bounds; again, this is because the post-TS probabilities have not changed by much. This stability of Mixup performance under temperature scaling is a possibly interesting direction for future exploration.

We also point out that while in our setting the post-TS ECE/ACE results are not substantially different, it has been observed in prior work (Guo et al., 2017) that temperature scaling can improve ECE significantly. We reconcile this with our results by noting that our experimental setup treats the case of ERM without additional regularization and trained to the interpolation regime of zero training error (since this is the focus of our theory). In prior work, the models considered were trained with several additional regularizations (data augmentation, weight decay, etc.), which will certainly have a notable impact on calibration performance.

Lastly, the results of Tables 6 to 8 also further corroborate our main theoretical ideas: even though temperature scaling can lead to improvements for the ERM models, the gap between ERM calibration performance and Mixup performance simply cannot be closed by temperature scaling alone.

| Dataset | Label Noise | ERM | ERM + TS | Mixup | Mixup + TS |
|---|---|---|---|---|---|
| CIFAR-10 | 0% | 23.2 ±1.4 | 22.6 ±0.6 | 7.1 ±1.0 | **6.7** ±0.5 |
| | 25% | 36.4 ±2.0 | 35.4 ±1.9 | 10.6 ±2.0 | **8.9** ±1.6 |
| | 50% | 56.4 ±1.0 | 55.4 ±1.1 | 17.6 ±5.7 | **16.4** ±5.1 |
| CIFAR-100 | 0% | 45.5 ±4.1 | 43.7 ±4.1 | 8.1 ±1.2 | **6.8** ±1.3 |
| | 25% | 51.2 ±1.6 | 48.9 ±1.6 | **6.3** ±1.6 | 6.4 ±1.5 |
| | 50% | 61.0 ±1.6 | 58.5 ±1.6 | 6.5 ±2.5 | **6.2** ±2.0 |
| SVHN | 0% | 7.1 ±0.5 | **7.0** ±0.5 | 9.2 ±0.8 | 7.8 ±0.6 |
| | 25% | 21.4 ±2.2 | 20.7 ±2.3 | **8.9** ±3.3 | 9.5 ±3.6 |
| | 50% | 48.8 ±1.7 | 48.0 ±1.8 | 13.1 ±2.6 | **12.0** ±2.3 |

Table 7: Mean ECE (multiplied by 100 and rounded to the nearest tenth) over 5 runs with 1 standard deviation error bounds on image classification benchmarks for TS and non-TS ERM and Mixup.

| Dataset | Label Noise | ERM | ERM + TS | Mixup | Mixup + TS |
|---|---|---|---|---|---|
| CIFAR-10 | 0% | 27.1 ±1.4 | 25.6 ±1.9 | **10.1** ±1.3 | 10.2 ±1.1 |
| | 25% | 29.7 ±0.9 | 29.1 ±1.1 | 12.2 ±3.3 | **9.3** ±1.8 |
| | 50% | 37.4 ±0.8 | 36.9 ±1.1 | **20.6** ±4.7 | 21.0 ±3.9 |
| CIFAR-100 | 0% | 38.3 ±2.3 | 37.0 ±2.5 | 10.0 ±2.0 | **8.2** ±1.5 |
| | 25% | 41.2 ±1.2 | 39.9 ±1.9 | **9.9** ±2.5 | 10.0 ±2.2 |
| | 50% | 47.1 ±1.4 | 46.1 ±2.2 | 12.7 ±5.5 | **11.8** ±3.2 |
| SVHN | 0% | 24.3 ±4.1 | 24.4 ±4.1 | **6.4** ±0.9 | 7.4 ±2.4 |
| | 25% | 21.7 ±1.8 | 20.5 ±1.9 | **9.4** ±1.4 | 9.9 ±1.2 |
| | 50% | 28.8 ±1.0 | 28.5 ±1.5 | **18.8** ±2.8 | 19.0 ±3.5 |

Table 8: Mean ACE (multiplied by 100 and rounded to the nearest tenth) over 5 runs with 1 standard deviation error bounds on image classification benchmarks for TS and non-TS ERM and Mixup.

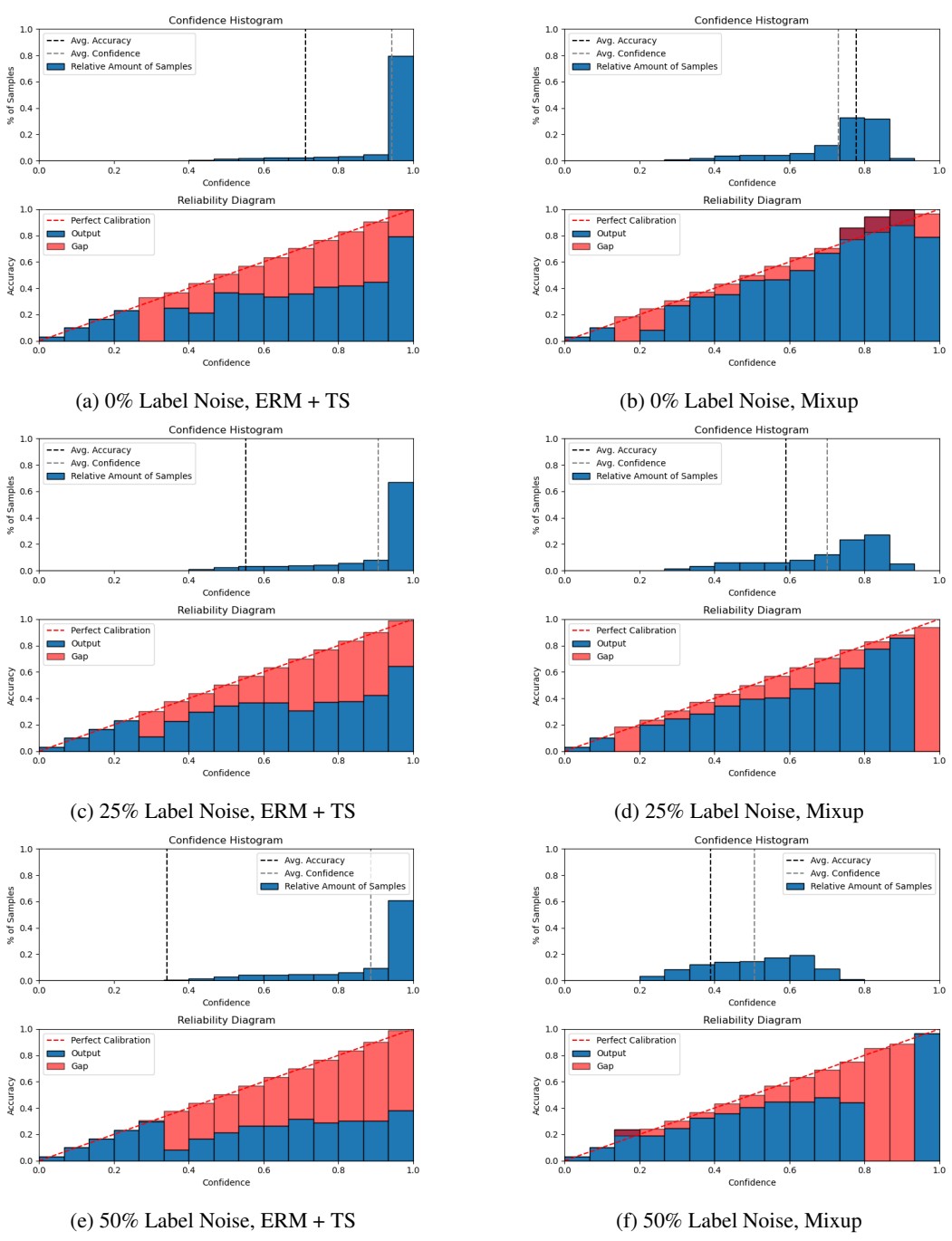

Figure 7: Confidence histograms and reliability diagrams for ERM + TS and Mixup models on CIFAR-10 at different levels of label noise.

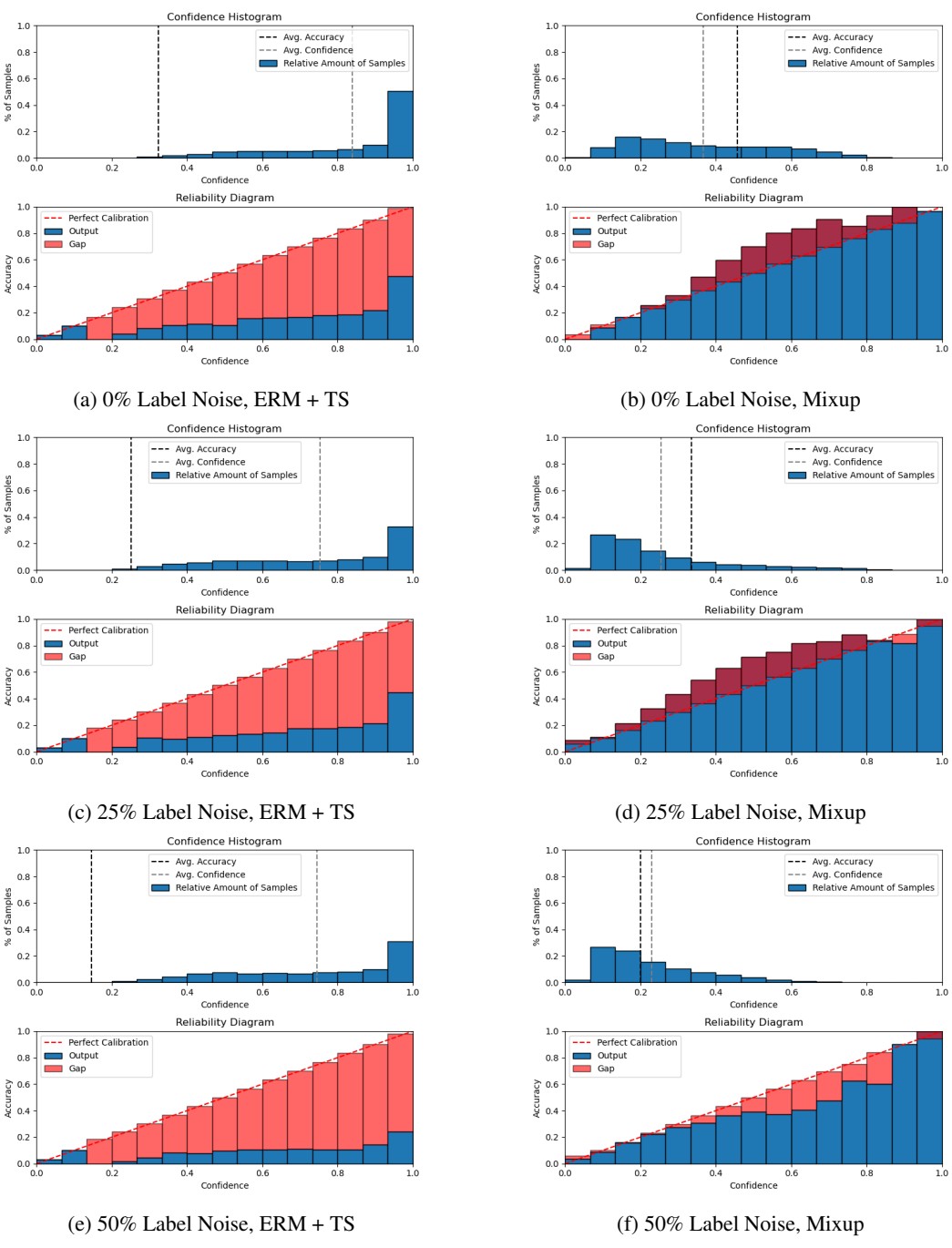

Figure 8: Confidence histograms and reliability diagrams for ERM + TS and Mixup models on CIFAR-100 at different levels of label noise.

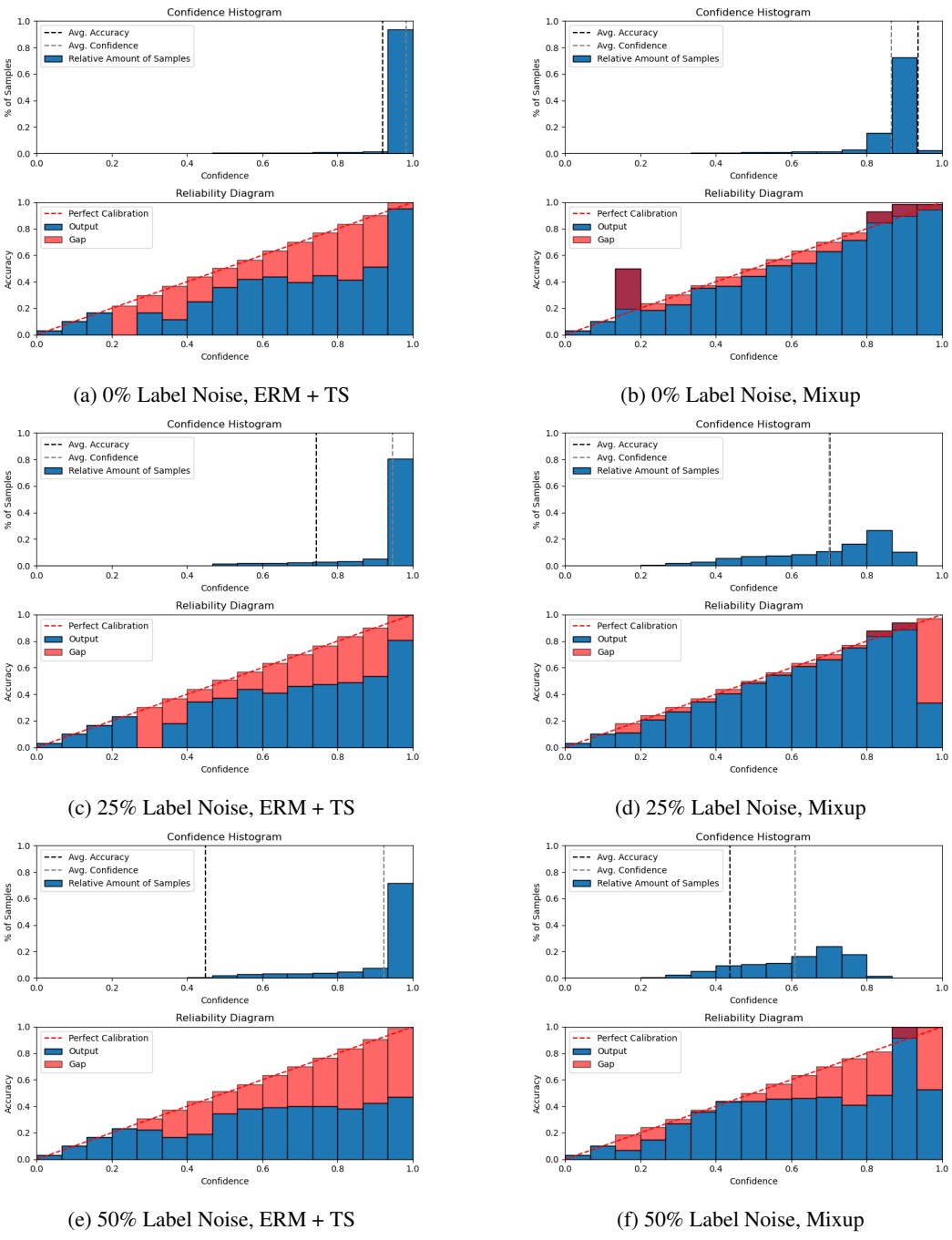

Figure 9: Confidence histograms and reliability diagrams for ERM + TS and Mixup models on SVHN at different levels of label noise.

