# OpenReview forum: "On the Limitations of Temperature Scaling for Distributions with Overlaps"
_ICLR.cc/2024/Conference — ICLR 2024 poster_

### Official Review · Reviewer_JXm6 · 2023-10-30

**Soundness:** 3 good
**Presentation:** 4 excellent
**Contribution:** 2 fair
**Rating:** 6
**Confidence:** 3

**Summary:**

This paper studies when and why temperature scaling may not work. This paper showed theoretically that when the supports of classes have overlaps, , the performance of temperature scaling degrades with the amount of overlap between classes, and asymptotically becomes no better than random when there are a large number of classes. This paper suggests that Mixup data augmentation technique can lead to reasonably good calibration performance, which are supported by the experiments conducted in the paper.

**Strengths:**

This paper identifies theoretical limitations of a widely used calibration technique, temperature scaling. The paper is technically solid and clearly written, with aligned theory and experiments.

Originality: The paper offers novel theoretical results on the inherent limitations of temperature scaling based on distributional assumptions. While temperature scaling is widely used, formal characterization of when it provably fails is new. The conditions identified are also intuitive and realistic.

Quality: The theoretical results are informative. [Note: I am not an expert in theory though, and I didn't check the proof]. The experiments cleanly validate the theory on both synthetic data and real image benchmarks. The proposed d-Mixup method is interesting. Overall the paper reflects quality research.

Clarity: The problem is motivated well and background provided. The writing clearly explains the theories, assumptions, experiments, and connections between them. Figures aid understanding. The paper is well organized.

Significance: Calibration is critical for uncertainty aware models, but little theory exists. This paper significantly advances understanding of an important technique. The insights on training procedures are impactful for future work.

**Weaknesses:**

1. The scope is limited to temperature scaling and Mixup. Discussing connections to other calibration methods could broaden impact.
2. It would be better to have more real data experiments. In the "IMAGE CLASSIFICATION BENCHMARKS", the overlap is introduced rather artificially.

**Questions:**

1. "We also trained d-Mixup models on the same data, but we found that the confidence regularization effect for d > 2 led to underconfidence on these datasets, so we report just the results for Mixup.": do we know why this may happen?
2. In the image experiments, for CIFAR-100, why having label noise makes NLL worse but ACE / ECE better? This makes the experiments less convincing if we don't have a solid explanations.

---

> ### Author Response · Authors · 2023-11-14
> **Response to Reviewer JXm6**
>
> We would like to thank Reviewer JXm6 for reviewing our paper, and we are grateful that they found our work to be clear and technically well-founded. We hope to address the raised weaknesses and questions below.
>
> 1. **Scope of discussed methods.** We definitely agree that it would be possible to extend the ideas from our theory to other data augmentation/training-time regularization approaches, and we have now emphasized this more in Section 1.1 (outline) in addition to stressing it in the conclusion.
>
> 2. **Real data experiments.** We also agree that having a more canonical "overlap" setting would be useful, however we felt that label noise was a natural way to introduce this overlap that also is readily encountered in practice (since samples are often mislabeled). Our results in Sections 5.2 and B.3 also include the 0\% label noise setting, which involves no artificial modification.
>
> 3. **Explaining regularization effect of $d$-Mixup.** When mixing several points with a uniform mixing distribution, mixtures in which one point gets a label very close to its original label (i.e. a mixing weight very close to 1) become less likely, and as a result we get even less spiky predictions (much closer to uniform probabilities).
>
> 4. **NLL gets worse but ECE/ACE get better.** It is possible to have very poor NLL while still having good (or even perfect) ECE, so worse NLL does not necessarily imply worse ECE performance. For example, a model that predicts uniformly at random can achieve zero ECE in the balanced class regime despite having poor NLL. Additionally, the ECE results in Table 4 for Mixup on CIFAR-100 are roughly within the 1 standard deviation error bounds of each other (so this does not represent a marked improvement), which we have also now pointed out in the surrounding discussion.
>
> Thank you again for your feedback and comments and we are happy to answer any further questions you may have.

---

### Official Review · Reviewer_dtwE · 2023-10-31

**Soundness:** 3 good
**Presentation:** 3 good
**Contribution:** 3 good
**Rating:** 6
**Confidence:** 3

**Summary:**

The paper demonstrated how temperature scaling has subpar performance in the case of overlapping classes, and proposed mixed up as an effective alternative to improve model calibration. The paper considers the ERM interpolator set of models, where there's clear separation between the prediction of top-class and the rest. In this case temperature scaling is failing to produce the desired behavior of equal prediction on the overlapping portion of the two classes. On the other hand, training with mixing loss is able to capture the overlapping behavior and significantly improve ECE.

**Strengths:**

- The paper is well written with clear presentation of references on background, assumption, and key results.
- There are theoretical results backing up the observations made in the toy examples and experiments.
- The few experiments shows good evidence supporting the conclusion that mixup is effective under the overlapping classes scenario.

**Weaknesses:**

- The mixing training introduces an extra degree of freedom (i.e. d-mixing). Based on table 2, we see that mixing is actually negative for NLL when classes are relatively separate, but only show performance improvement as the overlap increase. I also believe that the model performance would not be strictly better as we increase the degree of mixing, not to mention the additional computational complexity. Intuitively, the optimal d should have to do with the structure of overlapping in the dataset. I think it would be beneficial for the authors to have a more in-depth discussion on the choice of mixing in practice. Discounting the additional regularization effect, is it reasonable to only have the regular mixup when only two classes overlap at a time?

**Questions:**

- Does it make sense to generate additional classes for the overlapping case (y=1, y=2, y=1&2)? In that case would temperature scaling still work and what is the tradeoff here?
- Is ERM interpolator the best model class to capture the datasets with overlaps? The properties of the interpolator seem to be naturally mismatched.

---

> ### Author Response · Authors · 2023-11-14
> **Response to Reviewer dtwE**
>
> We would like to thank Reviewer dtwE for reviewing our paper, and we are happy to see that they found our work to be well-written and our claims to be well-evidenced. We hope to address the mentioned weaknesses and questions below.
>
> 1. **Amount of mixing in $d$-Mixup.** We definitely agree that the optimal mixing choice depends on the underlying data distribution being considered, and it is certainly true that mixing more points is not necessarily better (as we mention, this can lead to over-regularization of the predicted probabilities). However, it is tricky to analyze the optimal mixing choice in practice outside of treating it as an additional hyperparemeter. This is due to the fact that we do not know the underlying ground truth distribution - if we did have some way to quantify the amount of overlap in the ground truth, then it is certainly conceivable that we could develop theoretical guidelines for the optimal mixing number (although this would be separate from the theory in our work, which mixes $d + 1$ points since we do not make assumptions on the model class). Additionally, even when only two classes are overlapping there can still be significant benefit to mixing more than two points, which is demonstrated by the high overlap case in our experiments in Section 5.1 (4-Mixup performs the best in terms of NLL and ECE). The intuition here is again that the number of mixed points corresponds to the amount of regularization, and in the cases of high overlap we want to make sure the predicted probabilities are far away from being spiky (i.e. close to point masses).
>
> 2. **Generating additional class labels.** Generating synthetic labels as a means of regularization is definitely a useful idea, and in fact this is already done by methods such as label smoothing and Mixup (which drives the theory behind Mixup in our paper). The main difficulties with what you are suggesting (constructing a new class entirely) is the lack of knowledge of the ground truth distribution required to do so (that is, the input data does not specify which samples are in the overlapping part between class 1 \& 2). We need to specify some strategy for determining which samples get split off into the newly constructed class, and any such strategy will necessarily come with some baked-in biases about how we expect the ground truth distribution to behave. Naive strategies such as randomly assigning samples to the new class probably fall within the realm of existing label augmentation techniques, such as variants of label smoothing and Mixup.
>
> 3. **ERM interpolator drawbacks.** It is true that the properties of the ERM interpolators that we study are not ideal for calibration, which also drives the failure of temperature scaling in this case. However, we focus on this class of interpolating models as they are incredibly common in practice, and for these models we observe empirically the kind of assumptions we make in our theory (spiky probabilities that are well-separated). It is definitely possible that training ERM models with early stopping can provide sufficient regularization for better calibration (especially when combined with temperature scaling), which would be an interesting avenue to explore.
>
> If you have any further questions we are happy to answer them; thank you again for your useful feedback and remarks.

---

> ### Comment · Reviewer_dtwE · 2023-11-20
>
> Thank you very much for the response. This might be somewhat a naive question. Your response mentioned that we do not know the ground truth distribution and quantification of the overlap, but shouldn't we have the estimation from the empirical distribution through the training data and labels?

---

> > ### Author Response · Authors · 2023-11-20
> > **Follow-up Response to Reviewer dtwE**
> >
> > No problem, we hope the response adequately answered your concerns. Regarding your follow-up question, it is true that we obtain an estimate of $\mathbb{P}(Y \mid X)$ via training our model $g(X)$. However, this estimator is obtained from finitely many samples that are labeled as either one class or the other, and thus from an information-theoretic perspective does not necessarily provide any provably correct information about the overlaps in the ground truth distribution (without making assumptions on the structure of the ground truth distribution and the implicit biases of the trained model $g$). In general, we run into the same problem as before - we will need to specify some relabeling scheme of the data based off of $g$, which will necessarily introduce some biases about how we expect the ground truth distribution to behave.
> >
> > As a more concrete example, we can consider the (theoretical) case of what happens when we minimize the empirical negative log-likelihood (the interpolation regime) on data points $(x_1, y_1), ..., (x_n, y_n)$ over a sufficiently large model class. In this case, the learned model $g$ predicts $y_i$ with probability 1 for each $x_i$. Given this, it is not clear how to design a relabeling scheme based off the predictions of $g$ on the training data (since they will be identically 1). Even in the more practical setting where the model predictions are not identically 1 (but are very close to 1, as in the experiments in our paper), there is no guarantee that the predicted probabilities can be used to quantify some notion of overlap without making further assumptions on the relationship between model training and the ground truth distribution (the relative ordering of predicted probabilities may not be informative).
> >
> > We hope that further clarifies point 2 in our original response, and we are happy to answer any further follow-up questions.

---

### Official Review · Reviewer_UALj · 2023-11-03

**Soundness:** 4 excellent
**Presentation:** 4 excellent
**Contribution:** 3 good
**Rating:** 8
**Confidence:** 2

**Summary:**

This paper studies the limitations of the widely used temperature scaling for post-hoc uncertainty calibration. The authors find that under the specific assumption, i.e., the datasets follow a general set of distributions in which the supports of classes have overlaps, the temperature scaling method cannot perform well. Since the temperature scaling has been very successful in post-hoc calibration, this paper is interesting for pointing out its limitations and find under some conditions it provably fails to achieve good calibration. Furthermore, the authors find that the performance of temperature scaling degrades with the amount of overlap between classes, and asymptotically becomes no better than random when there are a large number of classes. This paper also studies a specific training-time calibration technique mixup and finds that it can lead to reasonably good calibration performance under the same conditions.

**Strengths:**

1. This paper is well motivated. The studied point is interesting. It is empirically found that temperature scaling is good for calibrating deep models. However, as pointed out in this paper, it may harm calibration under some conditions.

2. The empirical study shows very supportive results for the theoretic results. Both experiments on synthetic data and real-world data show positive results that temperature scaling cannot work under some conditions.

3. The writing and organization of this paper is very good.

**Weaknesses:**

1. It is commonly believed that temperature scaling is very effective for post-hoc calibration scaling, although it is found that this technique cannot be used for all the cases. Can you explain what causes this gap between your theoretic results and the commonly observed empirical success?

2. The main experimental results in tables only show the comparison between ERM+TS vs Mixup. I think that the results of ERM baseline and Mixup+TS should be presented at least. Moreover, is the results influenced by the training schemes used for training models (such as learning epochs, learning rate and regularization)?

**Questions:**

Please refer to Weakness section.

---

> ### Author Response · Authors · 2023-11-14
> **Response to Reviewer UALj**
>
> We would like to thank Reviewer UALj for taking the time to review our paper, and we are glad they found the work to be well-motivated and interesting. We hope to address their main questions below.
>
> 1. **Reconciling theory with common temperature scaling wisdom.** As you correctly point out, temperature scaling has been demonstrated repeatedly since [1] to be useful for calibration performance. Our theory does not contradict the *relative benefit* of temperature scaling (i.e. ERM + TS will almost certainly be better than ERM alone), but rather says that in *absolute* terms that there are classes of data distributions for which the improvement due to temperature scaling is simply not enough to achieve good calibration. This is made clearer by a new set of comparisons (Tables 6-8) that we have added as part of Appendix Section B.4, which show the relative gain from temperature scaling for both ERM and Mixup. As can be seen in this table, temperature scaling improves ERM calibration performance, but nowhere near enough to be even close to competitive with Mixup alone. Also worth pointing out is that in our setting, we do not train our ERM models with additional regularization (data augmentation, weight decay, etc.) as was done in prior work (so as to stay within the setting of our theory), and these almost certainly play a non-trivial role in how much temperature scaling can improve the base model (since the aforementioned modifications can lead to less spiky predictions and also better test accuracy).
>
> 2. **Non-temperature-scaling comparisons and effect of hyperparameters.** Thank you for the useful point; as mentioned above, we have added ERM vs ERM + TS and Mixup vs Mixup + TS comparisons to the Appendix as part of Section B.4. The main obversation from these comparisons is that while TS can significantly improve ERM performance (at least with respect to negative log-likelihood), it simply does not make a substantial difference with respect to the gap in performance between ERM and Mixup (exactly as suggested by our theory). With regards to the question about hyperparameters - calibration performance will certainly be affected heavily by hyperparameter tuning, with early stopping (among the modifications you mention) in particular having a significant effect (since this can prevent predicted probabilities from becoming too spiky for ERM). Our results are constrained to the common interpolation regime in which models are trained to have very low training loss, and in this regime we found that minor tweaks to hyperparameters such as learning rate did not have a significant effect (since we are training for a long enough time horizon). Additionally, as mentioned above, in this regime it is also harder to improve calibration performance since model predictions are even closer to point masses (which drives the failure of temperature scaling in our theoretical results).
>
> We are happy to answer any further questions you may have - thank you again for your review and helpful comments.
>
> [1] https://arxiv.org/abs/1706.04599

---

### Author Response · Authors · 2023-11-14
**Summary of Revision**

We would like to thank all of the reviewers for their helpful feedback and kind words regarding the contributions of our work. We have clarified parts of our paper to incorporate answers to reviewer questions. The main revision we have made is the addition of Section B.4, which now includes a comparison between ERM without temperature scaling and Mixup with temperature scaling to assess the relative impact of temperature scaling on the two approaches. We hope this adds more context to the results in the main paper, and we are happy to answer any further questions that any of the reviewers may have.

---

### Meta-Review · Area_Chair_WMd3 · 2023-12-06

**Metareview:**

This paper theoretically identifies a failure case of temperature scaling calibration: input distributions with “overlaps” (i.e. when p(y|x) is not deterministic for some regions of the input space). The authors also demonstrate that mixup is more efficacious in this setting at achieving calibration. Overall, the results are well presented, and an important contribution to the calibration literature. Empirical results validate the theory. I believe it will be of interest to the ICLR community and therefore recommend acceptance.

**Justification For Why Not Higher Score:**

Although all of the reviews were favorable, they were all low confidence (either due to a lack of familiarity with recent developments, or only a cursory glance over the theory). I read the paper and proofs myself to ensure that there were no major concerns, but - due to the lack of reviewer confidence - I do not feel comfortable recommending this paper beyond a poster.

**Justification For Why Not Lower Score:**

This paper is a novel and significant contribution. There are no major concerns about the theory or the results.

---

### Decision · Program_Chairs · 2024-01-16

Accept (poster)